

# Limitations of the Radon Tracer Method (RTM) to estimate regional Greenhouse Gases (GHG) emissions – a case study for methane in Heidelberg

5  Ingeborg Levin[1], Ute Karstens[2], Samuel Hammer[1,3], Julian DellaColetta[1,3], Fabian Maier[1,3], Maksym Gachkivskyi[1]

[1]Institut für Umweltphysik, Heidelberg University, INF 229, 69120 Heidelberg, Germany
[2]ICOS Carbon Portal, Lund University, Geocentrum II, Sölvegatan 12, 22362 Lund, Sweden
10  [3]ICOS Central Radiocarbon Laboratory, Heidelberg University, Berliner Straße 53, 69120 Heidelberg, Germany

*Correspondence to*: Ingeborg Levin (Ingeborg.Levin@iup.uni-heidelberg.de)



**Abstract.** Correlations of night-time atmospheric methane ($CH_4$) and $^{222}$Radon ($^{222}$Rn) observations in Heidelberg, Germany, were evaluated with the Radon Tracer Method (RTM) to estimate the trend of annual $CH_4$ emissions from 1996 – 2020 in the catchment area of the station. After an initial 30% decrease of emissions from 1996 to 2004, no further systematic trend but small inter-annual variations were observed thereafter. This is in accordance with the trend of emissions until 2010 reported by the EDGARv6.0 inventory for the surroundings of Heidelberg. We show that the reliability of total $CH_4$ emission estimates with the RTM critically depends on the accuracy and representativeness of the $^{222}$Rn exhalation rate from soils in the catchment area of the site. Simply using $^{222}$Rn fluxes as estimated by Karstens et al. (2015) could lead to biases in the estimated greenhouse gases (GHG) fluxes as large as a factor of two. RTM-based GHG flux estimates also depend on the parameters chosen for the night-time correlations of $CH_4$ and $^{222}$Rn, such as the night-time period for regressions as well as the $R^2$ cut-off value for the goodness of the fit. Quantitative comparison of total RTM-based top-down with bottom-up emission inventories requires representative high-resolution footprint modelling, particularly in polluted areas where $CH_4$ emissions show large heterogeneity. Even then, RTM-based estimates are likely biased low if point sources play a significant role in the station/observation footprint as their emissions are not captured by the RTM method. Long-term representative $^{222}$Rn flux observations in the catchment area of a station are indispensable in order to apply the RTM method for reliable quantitative flux estimations of GHG emissions from atmospheric observations.

## 1   Introduction

Monitoring the global distribution and trends of greenhouse gases (GHG) such as carbon dioxide ($CO_2$) and methane ($CH_4$) in marine background air dates back to the 1950s and 1980s, respectively (Brown and Keeling, 1965; Pales and Keeling, 1965; Blake and Rowland, 1988; Dlugokencky et al., 1994). With few exceptions, continuous continental GHG measurements started only in the 1990s, with a denser network established for $CH_4$ in the first decade of this century. In Europe, $CH_4$ observations are used in inverse (top-down, TD) modelling studies since 2009 to estimate the EU27&UK emissions of this potent GHG and its changes (Bergamaschi et al., 2009; 2018; Petrescu et al., 2021). Estimated fluxes were regularly compared to bottom-up (BU) emission inventories, based on reported national emissions, e.g. in the framework of the Paris Climate Accord (UNFCCC, 2015). But only the 2019 Refinement to the 2006 Guidelines of the UNFCCC reporting system (Witi and Romano, 2019) acknowledged the complementary capability offered by TD approaches for the reporting of GHG emissions.

A possibility to estimate continental GHG emissions on the local scale is the so-called Radon Tracer Method (RTM, Levin et al., 1999). The RTM uses the fact that the activity concentration of the natural short-lived radioactive noble gas $^{222}$Radon ($^{222}$Rn), which is emitted from continental soils but barely from ocean surfaces, is an excellent tracer for boundary layer



mixing processes (e.g. Servant et al., 1966; Dörr et al., 1983; Porstendörfer, 1994). $^{222}$Rn can be used as a measure of the

"continentality" of an air mass as its radioactive lifetime of about 5.5 days is long enough that $^{222}$Rn can accumulate in air masses residing over the continent. On the other hand, its lifetime is short enough that the $^{222}$Rn activity concentration exhibits a strong vertical decrease from elevated values in the continental boundary layer to small activity concentrations in the free troposphere (Liu et al., 1984). Similar to other gases, which have net sources close to the ground, $^{222}$Rn accumulates in a shallow (nocturnal) boundary layer when vertical mixing is suppressed. Therefore, if the exhalation rate of $^{222}$Rn from

the ground is known, the correlated increases of $^{222}$Rn and the gas in question (here $CH_4$) can be used to estimate the flux of this gas. In the Integrated Carbon Observation System Research Infrastructure (ICOS RI: https://www.icos-cp.eu/), atmospheric $^{222}$Rn observations are recommended to use this tracer for transport model validation but also to apply the RTM at ICOS atmosphere sites.

The Radon Tracer Method has been deployed in the past for greenhouse and other gases emission and sink estimates (Levin, 1984; Gaudry et al., 1990; Levin et al., 1999; 2011; Biraud et al., 2000; Schmidt et al., 2001; Hammer and Levin, 2009). In all these studies, the $^{222}$Rn flux from the soil has been assumed as spatially homogeneous and varying only slightly on the seasonal time scale. Recent research has, however, challenged this perception of a homogeneous and temporally almost constant flux. Several attempts to model $^{222}$Rn exhalation rates from European soils revealed rather large spatial variability

(Szegvary et al., 2009; Lopez-Coto et al., 2013; Karstens et al., 2015). The heterogeneity of $^{222}$Rn exhalation is caused by spatial differences in soil texture and soil $^{226}$Radium content, the precursor isotope of $^{222}$Rn. But even larger variations of soil $^{222}$Rn exhalation rate are due to temporal changes in soil moisture, which strongly influences diffusive transport of $^{222}$Rn in the soil air (e.g. Nazaroff, 1992). Soil moisture is, thus, the governing parameter for the observed seasonal variations of $^{222}$Rn exhalation (Jutzi, 2001; Schwingshackl, 2013; Karstens et al., 2015). Short-term varying soil moisture has its largest impact

on the $^{222}$Rn flux during the summer half-year, when missing precipitation over days or weeks can lead to changes in top soil moisture by more than a factor of two within a few days (e.g. Wollschläger et al., 2009). The basic assumption for estimating GHG fluxes with the classical RTM, i.e. a well-known and more or less constant $^{222}$Rn flux from the soil is, thus, more than questionable.

Based on these findings, the aim of this study is to re-assess the potential, but also the limitations of the RTM for local-to-regional scale GHG flux estimation, based on 20+ years of continuous atmospheric $CH_4$ and $^{222}$Rn daughter observations at the Heidelberg measurement site. Along with meteorological information, regional footprint analyses and model-based sensitivity experiments, we evaluate the influences of $^{222}$Rn and $CH_4$ flux variability in the Heidelberg catchment area on the observed night-time $CH_4$/$^{222}$Rn ratios and RTM-based $CH_4$ emission estimates. This concerns not only short-term day-to-day

variations, but also potential long-term changes of the $^{222}$Rn flux to be expected in view of an increasing frequency of summer droughts in Europe. Finally, we compare the RTM-based $CH_4$ emissions estimates for 1996-2020 and their inherent



uncertainties with bottom-up $CH_4$ emissions as reported in the EDGARv6.0 inventory (Crippa et al., 2021) for the model-estimated influence area around the Heidelberg measurement site.

## 2 Methods

### 2.1 Radon Tracer Method (RTM)

The basis of the Radon Tracer Method is the well-known observation that all trace gases with net positive emissions from continental surfaces accumulate in a stable nocturnal boundary layer. In a simple one-dimensional approach, the observed rate of concentration change ($dC_g(t)/dt$) at a fixed height within this layer depends on the mean flux density $j_g$ of the gas and on the actual boundary layer height ($H(t)$)

$$\frac{dC_g(t)}{dt} = \frac{j_g}{H(t)} \qquad (1).$$

Eq. (1) holds for all stable gases, and can be modified by including a decay term for short-lived (radioactive) gases like $^{222}$Rn (Schmidt et al., 2001), leading to Eq. (2):

$$\frac{dC_{Rn}(t)}{dt} = \frac{j_{Rn}}{H(t)} - \lambda_{Rn} \cdot C_{Rn}(t) \qquad (2).$$

Here $\lambda_{Rn}$ is the radioactive decay constant of $^{222}$Rn. The unknown (virtual) mixing layer height $H(t)$, considered to be the
same for $^{222}$Rn and the trace gas g, can be eliminated by combining Eqs. (1) and (2) and solving for the flux density $j_g$ of the trace gas g. In practice, when applying the RTM on a single night, we use measured finite concentration changes $\Delta C_g$ and $\Delta C_{Rn}$ instead of differentials, leading to the mean trace gas flux density $j_g$ during the observation period:

$$j_g = j_{Rn} \cdot \frac{\Delta C_g(t)}{\Delta C_{Rn}(t)} \left(1 + \frac{\lambda \cdot C_{Rn}(t)}{\Delta C_{Rn}(t)/\Delta t}\right)^{-1} \qquad (3).$$

Correction for the radioactive decay of $^{222}$Rn is taken care of by the term in brackets in Eq. (3). When applying the RTM
during a typical night-time inversion situation, lasting from late evening to early morning (i.e. less than 10 hours), the maximum change of $^{222}$Rn activity concentration due to radioactive decay is less than 10%. Contrary to earlier studies (Schmidt et al., 2001; Hammer and Levin, 2009) we neglect this effect in our evaluations and use instead Eq. (4) without the correction term:

$$j_g = j_{Rn} \cdot \frac{\Delta C_g(t)}{\Delta C_{Rn}(t)} \qquad (4).$$

The systematic bias towards higher estimated $CH_4/^{222}$Rn slopes, if radioactive decay is not corrected for, is estimated in a dedicated model experiment (Sec. 3.5).



One may argue that the simple one-dimensional model of the RTM is principally only applicable during inversion conditions with a stable or decreasing boundary layer height H; such situations occur mainly during summer nights. However, in this study we apply the RTM also for other meteorological night-time conditions, when the trace gases – in our case $CH_4$ and $^{222}Rn$ - change synchronously. This is justified as we assume that the measured air sample during night consists of two components, emissions from the ground with a certain $CH_4/^{222}Rn$ ratio and residual layer air that has a $CH_4/^{222}Rn$ ratio similar to that at the start of the night time observation period. While the local nocturnal boundary layer builds up, a residual layer is formed above this surface layer, which has a similar concentration as the well-mixed atmosphere in the late afternoon (Stull, 1998). We also included synoptic changes observed mainly during winter, as we assume that short-term trace gas changes, if large enough, are still mainly governed by recently added emissions from the regional catchment area.

The RTM approach implicitly assumes comparably homogenous spatial source distributions of $^{222}Rn$ and the trace gas. This means that it is well suited for homogeneous flux distributions, while trace gas plumes from point sources are not captured as they are not correlated with the area source-type fluxes of $^{222}Rn$. RTM-based emission estimates will therefore always underestimate real total GHG emissions in the catchment of a station if point source emissions are relevant. Further, as the footprint is not explicitly considered, the RTM (only) provides an (unknown) footprint-weighted average estimate of the trace gas flux. Consequently, without accompanying model simulations, which explicitly link footprints with the underlying emissions in the footprint area, it is not possible to quantitatively compare RTM-based TD fluxes with BU inventories, unless their emissions are very homogeneously distributed.

## 2.2      Heidelberg measurement site and methane sources in its catchment area

Heidelberg is a medium size city (ca. 160'000 inhabitants, 49.42°N, 8.67°E, 116 m a.s.l.) in south-west Germany, located at the outlet of the Neckar valley and extending into the densely populated upper Rhine valley (see map in Fig. 1). Continuous GHG and $^{222}Rn$ measurements are conducted on the University campus, with air sampling from the roof of the Institute of Environmental Physics building from about 30 m above ground level (a.g.l.). Depending on local wind direction, $CH_4$ concentrations are potentially influenced by local emissions from a close-by residential area and the Heidelberg city centre to the east. To the north of the University campus we find intensively managed agricultural land with some cattle breeding further away in the north-east. A large industrial area, Mannheim/Ludwigshafen (MA/LU) with chemical industry (BASF), solid waste landfills and waste water treatment facilities is located about 20 km to the north-west of Heidelberg. Further $CH_4$ hot spot emission areas, although much further away are larger cities like Karlsruhe, Heilbronn and the highly populated Rhein/Main area. The 2010 $CH_4$ emissions distribution from EDGARv6.0 (Crippa et al., 2021) in an area of about 150 km x 150 km with Heidelberg located in the centre, is displayed as gridded map in the left panel of Fig. 1. Here the MA/LU area sticks out as a hot spot with annual emissions of more than 0.05 kg $CH_4$ m$^{-2}$, i.e. more than a factor of 3-5 larger than mean emissions from any of the 0.1° x 0.1° pixels in the closer surroundings of Heidelberg.




The topography of the Rhine valley (≈ north - south) and the Neckar valley (east - west) influences the regional air flow, being dominated by southerly winds (Fig. 2); north-westerly winds from the MA/LU area are less frequent. Typical wind roses for the year 2015 (separated into daytime and nighttime hours) are displayed in the upper panels of Fig. 2. From these distributions we also see that the wind velocity (radius of the distributions) measured at 37 m a.g.l. on the roof of the

Institute's building lies most frequently between 2 and 4 m s$^{-1}$. We calculated nighttime and daytime only footprints and simulated preliminary $CH_4$ and $^{222}Rn$ concentrations for Heidelberg for selected years to determine the main influence area of our measurements. These footprint and concentration simulations are based on hourly runs with the Stochastic Time-Inverted Lagrangian Transport model STILT (Lin et al., 2003), that was implemented at the ICOS Carbon Portal (https://www.icos-cp.eu/about-stilt). Footprints estimate the main influence area for ground level emissions on the

concentrations measured in Heidelberg at 30 m a.g.l., which is approximately located in its centre. With a mean observed wind velocity of 3 m s$^{-1}$ (about 11 km per hour, Fig. 2), the approximate distance an air mass travels within the seven hours we use for the correlation of $CH_4$ and $^{222}Rn$ changes in the RTM, would then be ca. 75 km. This is why we chose to display in Fig. 1 the distribution of $CH_4$ emissions for a total area of 150 km x 150 km ("large" catchment area), being aware that strongest influences come from sources closer to the station (see aggregated footprints in Fig. 2). We thus also mark, by

black rectangle, a so-called "small" catchment area in the EDGARv6.0 $CH_4$ emissions map and also in the map of aggregated footprints in Fig. 2.

Long-term trends of total annual mean EDGARv6.0 emissions from 1995 to 2018 for the large 150 km x 150 km, the small (ca. 70 km x 70 km) and a third "intermediate" (110 km x 110 km) catchment area are displayed in Fig. 3. The 2010 mean

seasonal cycle of the large catchment area is shown on the right of the figure. For all three catchment areas, a significant decrease of about 30% is reported from 1995 to 2010. In the small catchment area this trend is interrupted in 2011 by an abrupt increase, which is associated to an increase in the "gas flaring and venting sector" (EDGAR sector: PRO, Janssens-Meanhout et al., 2019) in the pixel where BASF is located. The average fluxes in the larger catchment areas show similar abrupt increases in 2011, but smaller in size. After consulting the EDGAR team, it turned out that this abrupt increase is an

artefact caused by the introduction of a new proxy for the gas flaring and venting sector in 2011 (D. Guizzardi, pers. communication). Before 2011 mean $CH_4$ fluxes from the large catchment area are similar to those of the small catchment, while the intermediate catchment area generally shows only 80 – 85% of that mean flux. As expected for a highly populated and industrialised region, we see only a small seasonality in anthropogenic $CH_4$ emissions, originating from the seasonality in the sector "energy for buildings" (EDGAR sector: RCO).


As already mentioned in Sec. 2.1, given their predominant point source nature, it will not be possible to provide reliable information on the total $CH_4$ source strengths e.g. from MA/LU with the RTM, as this method is only applicable for area sources that are similarly homogeneously distributed as those of $^{222}Rn$ (Eq. 4). Potentially large contributions from industrial





point sources to the total flux will thus be missing in the RTM-based TD flux estimate so that results are likely biased low. As large point source emissions have to be reported directly to the European pollutant release and transfer (E-RPRT) register data base (https://prtr.eea.europa.eu/) by the facility, these bottom-up data are, however, likely much more accurate than any top-down estimate, as they are often based on direct measurements. But the more homogeneously distributed area sources dominating in the immediate neighbourhood of Heidelberg, such as energy for buildings, road transport, enteric fermentation and de-centralised waste management will probably be well represented in the RTM-based flux estimates. In the inventories these fluxes are associated with much larger uncertainties than those from point sources, and are thus a rewarding target for the RTM.

### 2.3 Radon exhalation rates in the Heidelberg catchment area

The most important pre-requisite to apply the Radon Tracer Method for quantitative GHGs flux estimates are representative $^{222}$Rn soil exhalation rates in the catchment area. The four panels on the left of Fig. 4 show the spatial distributions of $^{222}$Rn fluxes in the large ca. 150 km x 150 km catchment area of Heidelberg as estimated by Karstens et al. (2015) from bottom-up soil parameters and modelled soil moisture. The upper left panels show the estimated $^{222}$Rn fluxes for January and July based on the 2006-2010 soil moisture climatology from the ERA-Interim/Land model, while the lower left panels show the flux distributions using the GLDAS Noah soil moisture (averaged over 2006-2012) (https://doi.pangaea.de/10.1594/PANGAEA.854715). Large differences are seen between the models. Along the Rhine river in the north-west of Heidelberg (black dot in the centre of each map) where also a few excavated lakes are located, we find reduced $^{222}$Rn fluxes compared to the areas in the immediate surroundings of Heidelberg. This flux reduction is caused by the assumption of Karstens et al. (2015) that the low water table depth close to the rivers reduces mean $^{222}$Rn exhalation rates. As was shown and discussed by Karstens et al. (2015), the flux estimates based on the two soil moisture models show huge differences in their absolute values all over Europe. In the surroundings of Heidelberg these differences are larger than a factor of two throughout the year. But in both maps we see similar seasonal variations of the $^{222}$Rn flux, which are due to the seasonality of soil moisture with highest values in winter and dryer soils in summer and autumn. Note that in the STILT model runs discussed in Sec. 3.5 we use the average of both $^{222}$Rn flux maps, which we call "climatology".

In Heidelberg we are in the favourable situation that long-term observations of the $^{222}$Rn flux from soils have been conducted since the late 1980s (Dörr and Münnich, 1990; Schüßler, 1996). Jutzi (2001) has gathered these early data from five long-term measurement sites south of Heidelberg with different soil types to estimate mean seasonal cycles of the $^{222}$Rn flux. The data from three of these sites, i.e. those which have soil properties closest to the soil textures underlying the map of Karstens et al. (2015), are displayed in Fig. 4 (upper right panel). Measurements from the sandy soils of M1 and M3 have not been included as they are less representative for our catchment and showed annual mean $^{222}$Rn fluxes a factor of two smaller than at all other sites, which have been sampled in the last ten years in the surroundings of Heidelberg (Schwingshackl, 2013). The $^{222}$Rn flux measurements south of Heidelberg had also been used by Karstens et al. (2015), together with more recent



measurements from Schmithüsen (2012) and Schwingshackl (2013), conducted north of Heidelberg to evaluate their bottom-up process-based calculations of the $^{222}$Rn flux for the respective pixels. They reported significant differences in $^{222}$Rn flux when based on the different soil moisture models, ERA-Interim/Land or GLDAS-Noah LSM, but also between models and

observations (cf. their Figs. 6 and 7). Here we compare in Fig. 4 (upper right panel) both model estimates for the two pixels where the measurement sites south of Heidelberg are located with the observations from M2, M4 and M5. These measured $^{222}$Rn fluxes for sandy loam (M2) and loam (M4 and M5) lie in between the two model estimates, with the latter covering a range of (annual) mean $^{222}$Rn fluxes of more than a factor of two. Therefore, if no representative $^{222}$Rn flux observations are available at a monitoring site where the RTM shall be applied, depending on the soil moisture model we chose for the $^{222}$Rn

flux estimate, GHG emissions will differ by a factor of two or more. In addition, if the distribution of soil types is very heterogeneous, this will cause further uncertainty in individual RTM-based flux estimation. Based on the maps shown in Fig. 4 for the Heidelberg catchment areas (large or small), this heterogeneity of soil textures together with water table depth flux adjustment would contribute about 15-30% to the spatial variability of estimated night time $CH_4/^{222}$Rn ratios.

On the other hand, the upper right panel of Fig. 4 indicates, that the relative seasonality is similar in the two modelled as well as in the observed fluxes. This seasonality of ± (25-30) % will introduce a seasonality in RTM-based GHG fluxes and needs to be corrected in the final results. Normalised, to the respective annual means, measured and modelled seasonality of $^{222}$Rn fluxes in the two pixels south of Heidelberg were, thus, calculated and are shown in the lower right panel of Fig. 4. Here we also plotted the normalised average seasonality of monthly mean observed $^{222}$Rn fluxes at M2, M4 and M5. The seasonality

of this mean observed flux (dashed line in Fig. 4, lower right panel) is used to normalise the $CH_4/^{222}$Rn slopes of the individual night time correlations (Sec. 3.1). To finally estimate annual mean $CH_4$ fluxes with the Radon Tracer Method (Sec. 3.4) we will use the mean observed total flux at M2, M4 and M5 of 18.3±4.7 mBq m$^{-2}$ s$^{-1}$. The uncertainty of this observation-based mean flux represents the 1σ standard error of the mean at all three sites.

In Fig. 4 we present only monthly mean $^{222}$Rn fluxes and their spatial and temporal variability. However, we also expect variability of the $^{222}$Rn flux from day to day due to short-term soil moisture variations (Lehmann et al., 2000). In order to estimate this variability, we would need $^{222}$Rn flux data at higher temporal resolution. Such high-frequency data are, however, not available for the Heidelberg catchment area. We therefore estimated hypothetical daily mean $^{222}$Rn fluxes from soil moisture data at the long-term measurement site Grenzhof, which is located about 6 km to the west of the Heidelberg

monitoring station. Monthly mean soil moisture measurements from Grenzhof 2007 – 2008 had already been shown in Karstens et al. (2015) in their comparison with *monthly* mean modelled soil moisture data (see their Fig. 7d). Here we use the *daily* mean measurements of soil moisture and temperature in the upper 30 cm of the soil from Grenzhof (Wollschläger et al., 2009) and estimate daily mean hypothetical $^{222}$Rn fluxes for this site with the same methodology as used by Karstens et al. (2015). We assume a $^{222}$Rn source strength of the soil material of Q = 40 mBq m$^{-3}$ s$^{-1}$, chosen such that the annual mean



$^{222}$Rn flux for 2007 and 2008 fits the annual average observation-based flux value for the Heidelberg catchment area (18.3±4.7 mBq m$^{-2}$ s$^{-1}$). Details of the calculations are given in Appendix A; the results are displayed in Fig. A1.

As expected from the soil moisture variability (Fig. A1 upper panel) the short-term changes of the hypothetical $^{222}$Rn flux (Fig. A1 middle panel) are smallest during December to March, when soil moisture is at its maximum and much less variable than during spring, early summer and autumn. In these latter seasons, the day-to-day variability can reach up to ±30%. On average the day-to-day variability of the virtual $^{222}$Rn flux at Grenzhof was estimated to ±10 % (Fig. A1, lowest panel). Besides this short-term variability, we also observe a large difference of soil moisture in early summer between the two years: The rather wet June and July 2007 yield more than 30% lower $^{222}$Rn fluxes than estimated for June and July 2008. Early summer and autumn months' precipitation and thus soil moisture can vary strongly, causing potentially huge differences in the $^{222}$Rn flux from year to year. These short-term and inter-annual variations of the $^{222}$Rn exhalation rate will contribute to the day-to-day and inter-annual variability of night-time CH$_4$/$^{222}$Rn ratios. They increase the uncertainty of individual (e.g. monthly) RTM flux estimates and potentially their long-term trends. Note that the dry summers of the last decade in Europe (e.g. Hanel et al., 2018) are likely associated with higher $^{222}$Rn fluxes, at least in summer and autumn. If not accounted for, these $^{222}$Rn flux variations may lead to systematic biases in RTM-based emission estimates and their long-term trends.

## 2.4    CH$_4$ measurements

Air sampling from the roof of the Institute of Environmental Physics building (INF 229) for gas chromatographic (GC) analysis was performed via two separate intake lines, one in the south-eastern and one in the south-western corner of the roof. These two intake lines were installed to detect potential very local contamination by GHGs emissions from the air exhaust of the building or from other very close-by sources. Only during very few occasions data were manually rejected, if concentrations from the two intake lines showed a major deviation. In all such cases this deviation could be attributed to a problem with the intake system. Half hourly mean values of both intake lines were then calculated and used for further evaluation. Data from the years 1996-1998 stem from sampling at the old IUP building (INF 366), about 500 m to the west of the new building (INF 229). Also in these early years, air was collected from the roof of the building from approximately 25 m a.g.l.. The GC instrumentation was the same as in INF 229.

The combined Heidelberg gas chromatographic system (Combi-GC) was designed to simultaneously measure CO$_2$, CH$_4$, N$_2$O, SF$_6$, CO and H$_2$. It was optimised to measure ambient concentration levels for each trace gas with a temporal resolution of 5 min (Hammer et al., 2008). For CH$_4$ analysis, a HP5890II (Hewlett-Packard) GC equipped with a Flame Ionisation Detector (FID) was used. Ambient air was dried to a dew point of ca. -35°C before analysis. Methane mole fraction is referenced to the WMO X2004A mole fraction scale (Dlugokencky et al., 2005) with a precision of about ±3 ppb for individual measurements. A linear response of the FID was assumed over the whole range of ambient CH$_4$ mole fractions.




For details of the measurement technique, see Hammer et al. (2008). Since January 2018, a Picarro G2401 Cavity Ring-Down Spectroscopy (CRDS) gas analyser was used for $CH_4$ analysis. Air for this analyser is collected from the south-eastern

intake line with one-minute mean values stored and averaged to half-hourly values, following the procedures of the European ICOS atmosphere network (ICOS RI, 2020). The typical standard deviation of these half-hourly data as calculated from the 1-minute data is about ±2-10 ppb, depending on ambient air variability. As for the GC, CRDS measurements are reported on the WMO X2004A mole fraction scale.

### 2.5        Atmospheric [222]Radon and meteorological measurements

Atmospheric $^{222}Rn$ activity concentration is determined via its measured $^{214}Polonium$ daughter activity using the static filter method as described by Levin et al. (2002). Based on the results from a European-wide radon comparison study, which included parallel measurements of the Heidelberg monitor with a calibrated radon detector from ANSTO (Williams and Chambers, 2016; Griffiths et al., 2016), we applied a constant $^{222}Rn/^{214}Po$ disequilibrium correction factor to the data of 1.11, and report all data on the ANSTO scale, which turned out to be another factor of 1.11 higher than the original IUP

Heidelberg calibration (Schmithüsen et al., 2017). Depending on the activity concentration level, half-hourly $^{222}Rn$ activity concentration measurements in Heidelberg have a typical uncertainty of ±15%, including the uncertainty of all correction factors. The wind sensors are mounted on a mast on the southern side of the Institute's roof, at a height of 37 m a.g.l. Until 2011, wind speed was measured using a spherical cup anemometer and wind direction by a weather vane. From spring 2011 onwards, wind speed and wind direction is measured using a 2D sonic anemometer (Thiess, Germany). For both instrument

generations data was averaged to 5 min means.

### 3      Results

### 3.1      Estimating mean night-time $CH_4/^{222}Rn$ ratios from half hourly observations

For the period of 1996 to 2020 (except for 1999, when the Institute moved from INF 366 to INF 229 and no $CH_4$ observations are available), we calculated least squares fits of the half-hourly atmospheric $CH_4$ and $^{222}Rn$ observations from

21:00 h to 4:00 h CET in the next morning. To ensure that meaningful signals are evaluated, we set a lower limit of 1.5 Bq m$^{-3}$ for the $^{222}Rn$ range during the correlation period, which is about half of a typical mean range during all nights. In most years more than 45 nights were left, in which the correlation coefficient ($R^2$) of the night time $CH_4/^{222}Rn$ regressions was better or equal to 0.7. Anthropogenic $CH_4$ emissions in the Heidelberg catchment area have only a small seasonal variation of less than ±15 % (Crippa et al., 2021, and Fig. 3 right panel), and there are no wetlands with temperature-dependent

anaerobic $CH_4$ production in our region. However, the $^{222}Rn$ exhalation rate from soils has a pronounced seasonality. In our observations and also in both model estimates the $^{222}Rn$ flux during winter is up to 30 % lower than the annual average and it is up to 26% higher during late summer months (Fig. 4, lower right panel). This seasonality of the $^{222}Rn$ flux imposes a seasonality on the $CH_4/^{222}Rn$ ratios. We therefore normalised (de-seasonalised) all ratios on a monthly basis by





multiplication with a corresponding factor to the annual mean $^{222}$Rn flux. In the following we will first discuss these
normalised CH$_4$/$^{222}$Rn ratios and only in Sec. 3.5 RTM-based CH$_4$ fluxes are estimated. This intermediate step was taken
because of the large uncertainty of the *absolute* $^{222}$Rn flux in contrast to its much better defined seasonality (cf. Sec. 2.3 and
Fig. 4).

All selected normalised CH$_4$/$^{222}$Rn regression slopes with an R$^2 \geq 0.7$ are displayed in Fig. 5 upper panel. On average, more
than 80% of CH$_4$/$^{222}$Rn slopes vary between about 7 and 30 ppb (Bq m$^{-3}$)$^{-1}$. However, we also occasionally find slopes,
which are much larger than 40 ppb (Bq m$^{-3}$)$^{-1}$. In order to evaluate how sensitive CH$_4$/$^{222}$Rn slopes are on the selected night-
time interval chosen for the regressions, we also calculated slopes for an increased and a reduced time span, i.e. from 20:00 h
to 5:00 h and from 22:00 h to 3:00 h CET. The general shape of the distributions (frequency of positive outliers) is very
similar and also the overall means differ by only ±3 %. However, differences can be more than 15% in individual years. We
also evaluated how sensitive the annual mean slopes are to the threshold of correlation coefficient R$^2$. When selecting only
the nights where R$^2$ is equal or larger than 0.8, mean slopes are about 3% higher than when including all slopes with an R$^2 \geq$
0.7. Thus, a small bias may be introduced, depending on the choice of the night-time regression interval and also depending
on the requested goodness of correlation between CH$_4$ and $^{222}$Rn. It is also important to note that the number of nights with
R$^2 \geq 0.7$ increases systematically with the length of the tested regression time periods. The RTM is based on the co-variation
of trace gases and $^{222}$Rn through changing atmospheric mixing. Since there is no causal correlation between the emission
processes of the two gases, their different spatial source heterogeneity in combination with changing footprints leads to a
reduced number of valid correlations with a shorter observation period. In contrast, more extended regression periods with
variable footprints increase the probability of averaging across spatial heterogeneity of emissions.

Interestingly, mean slopes are only about 3% different (larger) if only values obtained for situations when both
concentrations increase are included, compared to when we also include the about 20% situations when both gases show a
decrease between the start and the end of the regression interval. This finding may be a special characteristic of our sampling
site, where the air intake is only at 30 m a.g.l. During very stable situations and calm winds the air intake can obviously be
either below or above the local surface inversion (if this is around 30 m), which results in very abrupt but synchronous
changes of both gases in some nights. As mentioned in Sec. 2.1 we can describe this as a case where two air mass
components, i.e. one enriched by emissions from ground level sources with a well-defined CH$_4$/$^{222}$Rn ratio and another,
cleaner, component from the residual layer that has a CH$_4$/$^{222}$Rn ratio similar to that during well-mixed situations in the
afternoon before. These two components are mixed at various ratios. In such a situation all measured CH$_4$/$^{222}$Rn ratios lie on
one mixing line, which corresponds to the regression line in our approach. With this picture in mind, it becomes immediately
clear that in Eqs. (1) and (2) (Sec. 2.1), besides the concentrations of CH$_4$ and $^{222}$Rn, also the mixing height H(t) may vary
temporally and does not need to be constant during a single night to apply the RTM. We, thus, kept all nights when CH$_4$ and
$^{222}$Rn are well correlated for calculating annual means and further evaluating the slopes.





## 3.2 Relating CH₄/²²²Rn slopes to influence areas

The $CH_4/^{222}Rn$ slopes displayed in Fig. 5 show large variability, and we wondered, if this variability can be explained by
spatial variations in the $CH_4$ emissions, and if yes, if we can associate e.g. the high slopes to one of the hot spot emission areas in the footprint of Heidelberg. We, therefore, evaluated the air mass influence based on local wind data for all nights when we obtained good ($R^2 \geq 0.7$) correlation between $CH_4$ and $^{222}Rn$. Let us assume that the $^{222}Rn$ flux is spatially homogeneous; then we would expect higher slopes if the air mass origin is from the north-westerly or westerly sectors where the large $CH_4$ emitters from MA/LU are located (Fig. 1). Figure 6 shows in the first column polar plots of wind direction
(angle) and speed (radius axis) with the value of the corresponding slopes color-coded (i.e. larger slopes plotted in darker red colours). Note that we use here the original 5-minute mean values of wind speed and direction, together with the mean slope during the entire night (7 hours). Each polar plot shows the distribution for all selected nights of the entire year (2016, 2017 and 2018 as typical examples from the later years of our record); the colour-coded segments represent annual mean values of all slopes where a five-minute value fell into the respective wind rose segment. The second column of Fig. 6 shows the
frequency distribution of the wind during all selected nights, while the third column shows the distribution during all nights in the respective year (21:00 h – 04:00 h).

The frequency distributions of 2016 and 2017 indeed show higher average slopes when the wind comes from north-westerly directions, but in 2018 high slopes are also associated to the northern or north-eastern wind direction. Interestingly, the
easterly and south-easterly sectors show average slopes that are often smaller than about 20 ppb (Bq m⁻³)⁻¹. This is a wind sector where also EDGARv6.0 generally reports lower than average emissions (Fig. 1). A problem with this analysis is that during low wind speed, the wind direction is not well defined and may change by (more than) 180° within a single night. The measured air would then be influenced by emissions from various sectors with different $CH_4$ emissions. This could smooth out an otherwise clear association of slopes to certain wind sectors. Also, low wind speed situations are more frequent during
stable nights (as indicated for the selected nights in Fig. 6 middle panel) with a shallow boundary layer and large nocturnal increases of $CH_4$ and $^{222}Rn$, i.e. nights with good correlation between the two gases and where the RTM can be principally applied. We should also keep in mind that part of the high emissions in the MA/LU hotspot area are probably from point sources that will not be captured by the RTM. Also the frequency distribution of wind directions generally (for all nights) favours more southerly and south-easterly winds, which reduces the likelihood to monitor the high $CH_4$ emissions from the
MA/LU area. Nevertheless, can we roughly separate influence areas, which, on an annual mean basis, differ in their mean slopes by more than a factor of two. This indicates that a large share of the variability of slopes (Fig. 5) is caused by the heterogeneity of $CH_4$ emissions around Heidelberg.



### 3.3 The influence of $^{222}$Rn flux variability on the variability of CH$_4$/$^{222}$Rn slopes

Besides the heterogeneous distribution of CH$_4$ emissions in the Heidelberg catchment, we expect part of the variability in the CH$_4$/$^{222}$Rn slopes to be also due to variations of the spatial distribution of the $^{222}$Rn exhalation rate. Figure 4 shows the spatial $^{222}$Rn flux distribution for the large Heidelberg influence area in January and July for both soil moisture models. Although mean fluxes from the two different soil moisture models differ by more than a factor of two, the spatial variability within one map varies by only ±(15-25)% within the large catchment and slightly more in the small 70 km x 70 km

catchment area. Therefore, the spatial variability of the $^{222}$Rn flux probably contributes much less to the variability of slopes than that of the CH$_4$ flux (see also Sec. 3.5 where we investigate the contributions of CH$_4$ versus $^{222}$Rn flux heterogeneity on modelled CH$_4$/$^{222}$Rn slopes). Also the short-term day-to-day variability of the estimated "hypothetical" $^{222}$Rn flux, as elaborated in Appendix A and displayed in Fig. A1 for the years 2007 and 2008, may contribute to the variability of slopes. The hypothetical daily flux estimates, which are based on the measured daily mean soil moistures, show a mean day-to-day

variability of ±10%, but during early summer 2007, and likely also in other years, particularly during spring and autumn, short-term deviations from monthly mean fluxes can be as large as 30%. However, these deviations are still too small to explain a major share of the observed slope variability displayed in Fig. 5.

### 3.4 Estimating CH$_4$ fluxes with the RTM and comparison with EDGARv6.0 emission trends

As shown in the previous section, the spatial variability of CH$_4$ emissions and, to some extent, also the spatial and temporal variations of the $^{222}$Rn flux in the catchment area of Heidelberg are large and make reliable estimates of RTM-based CH$_4$ emissions from selected sectors (e.g. of industrial processes in MA/LU) or for individual short periods highly uncertain. But we can estimate average CH$_4$ emissions from the footprint of the station. As a first attempt to apply the RTM we use the observation-based $^{222}$Rn flux, which was estimated as the mean of our measurements at M2, M4 and M5 to 18.3±4.7 Bq m$^{-2}$

s$^{-1}$ (Sec. 2.3). The corresponding CH$_4$ flux it is plotted as black histogram in Fig. 7. The uncertainty of the absolute RTM-based CH$_4$ fluxes is dominated by the uncertainty of the mean $^{222}$Rn flux and is exemplarily plotted as black error bars for the first and last year of observations. A significant decrease of emissions by about 30% is observed from 1995 until about 2004. This decrease is in agreement with the trend of bottom-up EDGARv6.0 emissions from 1995 – 2010 calculated for all three catchment areas in Fig. 3. However, while EDGARv6.0 emissions show a further decrease after 2004, our RTM-based

estimates are more or less constant after 2004, showing an inter-annual variability of less than ±10%.

In Fig. 7 we also included the range of CH$_4$ emissions we would estimate when using the mean $^{222}$Rn flux from the maps by Karstens et al. (2015). For this estimate we used the mean $^{222}$Rn fluxes from the small catchment area. As expected from the huge difference in $^{222}$Rn fluxes between the two soil moisture models (Fig. 4), possible RTM-based CH$_4$ emission estimates

would cover a range of more than a factor of two (indicated in Fig. 7 by the coloured area). Using the *mean* $^{222}$Rn flux from





both model estimates, i.e. the climatology, would – accidentally - yield a similar (ca. 10% lower) RTM-based $CH_4$ flux as when using the observation-based $^{222}Rn$ flux for the Heidelberg catchment.

### 3.5 Comparing the observation-based RTM results with the RTM application on preliminary STILT $CH_4$ and $^{222}Rn$ simulations

One important shortcoming of RTM-based GHG flux estimates is the lack of information on the actual influence area for which the estimated flux is representative. In Sec. 2.2 and Fig. 2 we could only roughly localise the large ca. 150 km x 150 km catchment area for Heidelberg, contributing most of the source influence on the nighttime concentration changes within the 7 hours used for the RTM-based flux estimates. Quantitative comparison with bottom-up emission inventories, however, requires actual weighting of the influence area, in particular if the distribution of the GHG emissions is as heterogeneous as

in the Heidelberg surroundings. This weighting can be achieved with regional transport model simulations. For the following STILT model estimates the footprints were mapped on a 1/12° latitude x 1/8° longitude grid and were coupled (offline) to the EDGARv6.0 emission inventory (Crippa et al., 2021) for $CH_4$ concentration estimation, neglecting seasonality of emissions. We also simulated atmospheric $^{222}Rn$ activity concentrations based on the two $^{222}Rn$ flux maps of Karstens et al. (2015) (the average climatology of ERA/Interim-Land and Noah GLDAS was used for the simulations). The modelled

regional concentration components represent only the influence from surface fluxes inside the model domain (covering the greater part of Europe, i.e. an area much larger than the large catchment area defined in Sec. 2.2). The background concentrations for $CH_4$ and $^{222}Rn$ outside our modelling domain have been neglected as we are here only interested in night-time changes of both trace gases. We then applied the RTM also on these preliminary model results and compared the slopes and their typical distribution to those from the observations. Comparing modelled with observed slopes rather than absolute

concentrations has the advantage that incorrect parameterisation of the nighttime boundary layer height by the model partly cancels, while the relative footprint area weighting may still be reliable, even for nighttime simulations.

Figure 8 shows the normalised observed and modelled $CH_4/^{222}Rn$ slopes in Heidelberg for the years 2007 – 2010 and their distributions. We did run the STILT model also for 2011, but due to the error in the EDGARv6.0 emissions from 2011

onwards, we used the results only as a sensitivity test (see below). Although we use the same selection criteria for the modelled concentration regressions as for the observations, the number of nights with good correlations of $CH_4$ and $^{222}Rn$ is about five times higher than for the observations. Note that we do not want to compare here modelled with observed slopes of individual nights, e.g. in a scatter plot, because we are mainly interested to compare mean values (to further translate them into mean emission rates as displayed in Fig. 7) and their distributions. In the model-based slopes we find a number of very

high values, which we do not see in 2007 – 2010 in the observed slopes. We can clearly identify these high modelled slopes as being associated with north-westerly winds and thus as strong influence from hot-spot $CH_4$ emissions in these situations. Although the hot-spots in reality have most probably very localised emissions and are not captured by the RTM in the real world, in the model these emissions are distributed over the area of the entire about 10 km x 10 km wide pixel, so that during





stable winds good correlations between $^{222}$Rn and $CH_4$ may occur over an entire night, and very high $CH_4/^{222}$Rn ratios can be
obtained. This finding is confirmed by STILT model results for the year 2011, where $CH_4$ emissions in EDGARv6.0 are
more than doubled in the MA/LU pixel. In this year we find a larger number of high slopes than in the years 2007 – 2010,
some of them exceeding 100 ppb $(Bq\ m^{-3})^{-1}$.

If we exclude the three outliers above 70 ppb $(Bq\ m^{-3})^{-1}$ in 2008 and 2009 in the averaging of the modelled slopes, we obtain
rather good agreement with the mean observed slopes (i.e. observations = $(15.6\pm7.9)$ ppb $(Bq\ m^{-3})^{-1}$; model = $(16.7\pm8.5)$ ppb
$(Bq\ m^{-3})^{-1}$). Also the relative variability is then very similar in the modelled compared to the observed slopes, i.e. 50% vs.
52%. This justifies quantitative comparison between model results and observations. However, even under the assumption
that the modelled footprint area is correct, we are still not able to quantitatively validate EDGARv6.0 emission estimates
through comparison between model and observations as long as we do not know the true $^{222}$Rn flux in this footprint area. But
we can go one step further and normalise the model results to the same $^{222}$Rn flux as we believe is the best estimate for the
Heidelberg catchment area based on observations. The model simulations were based on the $^{222}$Rn flux climatology of
Karstens et al. (2015), which give an annual mean flux averaged over the small footprint area of 16.7 mBq $m^{-2}\ s^{-1}$ (the mean
flux in the large catchment would be 2.5% lower). Normalisation then increases the mean modelled slopes by a factor of
18.3/16.7, leading to an over-estimation of the modelled slopes compared to the observations by a factor of
model/observation = 16.7*18.3/16.7/15.6 = 1.17. The uncertainty of this result would be about 25%, i.e. the estimated
uncertainty of the mean observation-based $^{222}$Rn flux. Within this uncertainty we could come to the conclusion that
EDGARv6.0 emissions in the Heidelberg footprint area would be slightly over-estimated by $(17\pm25)$ %. However, we must
not forget that the observation-based RTM results (and, to some extent, also the STILT-based results) are biased low because
we do not (or only partly) catch emissions from very localised $CH_4$ sources. How big the respective biases are, is hard to
quantify; it would require a dedicated sensitivity study with a realistic very high-resolution transport model and an emission
inventory that separates area and point source emissions.

We further used STILT model simulation experiments to investigate the sole influence of (1) $CH_4$ flux heterogeneity, (2)
$^{222}$Rn flux heterogeneity and (3) neglecting radioactive decay of $^{222}$Rn in the calculation of $CH_4/^{222}$Rn slopes in Heidelberg.
For these experiments we compared the standard model results with those where we used (1) a constant $CH_4$ source
distribution, (2) a constant $^{222}$Rn flux and (3) treated $^{222}$Rn as a stable tracer. Experiments (1) and (2) confirmed that most of
the variability of $CH_4/^{222}$Rn slopes in Heidelberg is due to the heterogeneity of the $CH_4$ source distribution. Keeping $^{222}$Rn
fluxes constant had no significant influence on the standard deviation of the $CH_4/^{222}$Rn slopes, however, spatially
homogeneous $CH_4$ emissions reduced the variability of the slopes from about 50% to less than 20%. When treating $^{222}$Rn as
a stable tracer in the model, mean slopes were 7% lower than in the run, which included radioactive decay in the modelled
$^{222}$Rn activity concentration. This means that both, modelled and observed slopes need to be corrected downwards by 7%.



This has, however, no influence on our finding that EDGARv6.0 emissions in the Heidelberg catchment may be (17±25) % too high.

## 4    Discussion

### 465    4.1    How reliable can RTM-based GHGs flux estimates be?

The Radon Tracer Method is a purely observation-based method to estimate nighttime fluxes from homogeneously distributed ground level sources of trace gases. Its application is simple; in principle, it does not require sophisticated atmospheric transport modelling. Depending on the height above ground level of co-located $^{222}$Rn and trace gas observations, RTM-estimated fluxes can be representative for an area of several hundred square-kilometres. However, the

exact area for which the estimated mean nighttime flux is representative must be estimated separately, e.g. by footprint modelling. The accuracy of the RTM-based trace gas flux estimates is almost solely determined by the exact knowledge of the $^{222}$Rn exhalation rate from the soils in the catchment area of the atmospheric station. Still, even if the absolute $^{222}$Rn exhalation rate is not well known, and with that the absolute trace gas flux, the RTM can provide validation of long-term trace gas emission trends, for example of GHG emission reductions. This, however, requires that the $^{222}$Rn flux does not

show a systematic long-term trend, which, for example, may be caused by long-term changes of soil moisture in the catchment area of the measurement site. Also the mean footprint should not show a systematic trend, e.g. due to climate-driven changes of local transport patterns. This is particularly important if $^{222}$Rn and/or trace gas emissions show large spatial heterogeneity in the footprint.

The RTM-based $CH_4$ emission *trend* calculated from Heidelberg observations is in good agreement with the *trend* of the EDGARv6.0 bottom-up inventory data. However, after 2004 our observations do not show a further decrease, contrary to the values reported by EDGARv6.0. Comparison of *absolute* emissions is, however, difficult as point source emissions are not captured by the RTM; therefore, our RTM-based fluxes are biased low. As we rely on modelled footprints for a quantitative comparison of RTM-based top-down fluxes with inventory-based bottom-up emission estimates, it will depend on the share

of point source emissions how reliably we can compare observed with modelled slopes. Due to the coarse grid of the STILT model we used in this study and the coarse resolution of the inventory, point source emissions were distributed over 10 km x 10 km grid areas. This resulted in a larger number of high slopes in the model results compared to observations if the air mass came from the MA/LU hot spot emissions area. Modelling $CH_4$ and $^{222}$Rn with a higher resolution model and emission inventory could improve comparability of model results and observations, and therewith help quantifying the bias in

observation-based RTM results caused by point source emissions in a particular setting.

Large potential biases in observation- and model-based RTM flux estimates are introduced by the uncertainty of the $^{222}$Rn flux in the catchment area. For the Heidelberg catchment, the uncertainty of 25% for the mean $^{222}$Rn flux is probably an

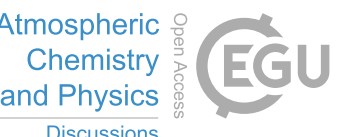

upper limit, because soil texture and [226]Radium content of the soils in the catchment of our station show only small

variability (<10%) (Schwingshackl, 2013; Karstens et al., 2015). But we would need more systematic and representative [222]Rn flux observations, also at larger distances from Heidelberg, to estimate a more accurate mean observation-based flux with smaller uncertainty range.

On the other hand, we want to emphasise that comparing simulated mean nighttime $CH_4$/[222]Rn slopes with observed slopes

could be a more accurate method to evaluate bottom-up emissions than directly comparing simulated and observed nighttime $CH_4$ concentrations or using model inversions of nighttime data to optimise $CH_4$ fluxes. This problem is certainly less serious if only daytime observations are used in the inversions. However, the about five-fold larger surface influences (sensitivity) during night than during day (Fig. 2) may help improving top-down results. The normalisation of modelled nighttime $CH_4$ with modelled [222]Rn largely eliminates errors in model transport, such as e.g. deficiencies in the

parameterisation of the nocturnal boundary layer height, but also in this approach the final outcome and its significance depend on the correctness of the underlying [222]Rn exhalation rate. This exhalation rate can easily have larger uncertainties than the GHG emission inventory we target to evaluate. For example, for Europe, different bottom-up $CH_4$ emission inventories agree to within 10% or better (e.g. Petrescu et al., 2021). It is still likely that the uncertainty of BU GHG fluxes in a smaller area, that have been disaggregated from national totals, and thus depend on generalised assumptions about

emission factors and proxies for the different sectors, are much larger than these 10%, or may even have flaws (see Sec. 2.2 and Fig. 3).

It should, perhaps, also be noted that our Heidelberg site may be a special case with advantages and disadvantages to apply the RTM. First, we have conducted the long-term observations with the same instrumentation, except for $CH_4$ in the last

three years. More importantly, the air intake at about 30 m a.g.l. may be favourable for RTM applications, as it frequently lies in the nocturnal surface layer, which implies that we observe sufficiently large nighttime increases of both gases to obtain good correlations. Nevertheless, at this height above ground we monitor a footprint that is large enough to not only being influenced by very local emissions. A major advantage for estimating potentially accurate $CH_4$ fluxes were long-term observations of the [222]Rn exhalation rate and its seasonality from typical soil types around the station. This made the results

presented here fully independent from modelled soil moisture-based [222]Rn flux estimation. If we had to solely rely on modelled [222]Rn fluxes, e.g. from Karstens et al. (2015) the uncertainty range of RTM-based estimates would have been as large as a factor of two (Fig. 7, coloured area). The largest disadvantage of our setting is, however, that $CH_4$ emissions in our catchment area are very heterogeneous and contain point sources, which cannot be evaluated with the RTM. Therefore, observation-based but also STILT-based $CH_4$ flux estimates are biased low to a currently unquantifiable extent.


There are a number of other issues that need to be kept in mind when applying the RTM: It is important to carefully evaluate what the most appropriate night time period is to calculate representative trace gas fluxes. We investigated this parameter for





Heidelberg and found on average about 3% smaller $CH_4$ fluxes when extending the regression period from 7 to 9 hours and 3% higher fluxes when reducing it to 5 hours. But for individual years mean slopes showed differences larger than 10%

when changing the length of the regression period. Also, in these scenarios the number of nights with good correlation (i.e. $R^2 \geq 0.7$) decreased significantly when the correlation period was shortened to 5 hours or even less. The heterogeneity of $CH_4$ emissions in the Heidelberg catchment area may have contributed to this effect, as we often have very variable wind directions during stable nights, and changes in the $CH_4/^{222}Rn$ slopes may lead to bad correlations if only a smaller number of data points are correlated. Also increasing the quality of the regression from $R^2 \geq 0.7$ to $R^2 \geq 0.8$ led to an increase of the

mean slope (here by 3% on average). As the average correlation coefficient did not change when changing the regression period and selecting only nights with $R^2 \geq 0.7$, we finally decided to fix this period to those 7 hours, which always, during winter and summer fall into dark night time (i.e. 21:00 h – 4:00 h CET). However, we have to admit that this decision was made in a rather subjective way.

### 4.2 Would reliable RTM-based GHG flux estimates be possible at ICOS stations?

At many stations in the ICOS atmosphere network continuous $^{222}Rn$ observations are conducted, however, almost no systematic $^{222}Rn$ flux observations exist close to these stations. This is a serious deficiency if the RTM shall be routinely applied in this network for top-down GHGs flux estimation. Even if these measurements may be introduced in the future, they need to be conducted at a number of representative soils in the catchment area and over a longer time period. We could show that the day-to-day variability of the $^{222}Rn$ exhalation rate can be large (Fig. A1). Also inter-annual variations of soil

moisture due to variations in seasonal precipitation ask for systematic long-term $^{222}Rn$ flux measurements to allow for representative estimates of the mean flux and its typical seasonality. A second problem to reliably apply the RTM at ICOS stations may be the relatively high air intake for $^{222}Rn$ (generally > 100m a.g.l.). Nighttime increases of soil-borne trace gases are much smaller at these heights than at 30 m, and the layer with the air intake may be decoupled from ground level emissions. This increases the catchment area of the station with potentially more heterogeneous and possibly less well-

defined $^{222}Rn$ fluxes.

However, we could show in our study that the long-term trends of RTM- and inventory-based emission estimates did not significantly deviate from each other. Monitoring potential trends of GHG fluxes is an important task of ICOS and could very well contribute to the regular stocktakes under the UNFCCC accord (UNFCCC, 2015), providing independent

validation of reported trends. Still, this would require confidence that $^{222}Rn$ fluxes have not changed over the monitoring period.

### 4.3 Could a better $^{222}Rn$ flux map help to improve RTM-based GHG flux estimates?

As was shown in Fig. 4, the current $^{222}Rn$ flux maps from Karstens et al. (2015) show huge differences depending on the soil moisture model that was used. In the case of Heidelberg, a simple averaging of these two model estimates (what we called





climatology) would have fit rather well to the observations (the average $^{222}$Rn flux for the Heidelberg catchment area would then be between 16.3 mBq m$^{-2}$ s$^{-1}$ and 16.7 mBq m$^{-2}$ s$^{-1}$, compared to the observation-based flux of 18.3±4.7 mBq m$^{-2}$ s$^{-1}$). Averaging both estimates would thus have been a tempting solution for the Heidelberg catchment if no observations had been available. But would averaging both maps yield reliable estimates of the $^{222}$Rn flux also at other sites in Europe? As was shown by Karstens et al. (2015), it is not obvious that one or the other soil moisture model or the average of both models

would fit observed $^{222}$Rn fluxes best. There is some indication that the ERA/Interim-Land-based fluxes are generally underestimating observations (Karstens et al., 2015, Fig. 8). Today, improved so-called third generation land reanalysis models are available (see Li et al., 2020, for an overview). Soil moisture estimates from these third generation models have been compared to observations and it turned out that "the European Centre for Medium-Range Weather Forecasts ERA5 model (Hersbach et al., 2018) shows higher skills than the other four products and a significant improvement over its

predecessor" (Li et al., 2020). However, although the ERA5 results give realistic variability, they often show systematically higher soil moisture than the observations. In order to use these new reanalysis data, which have the advantage that they are available now at much higher temporal and spatial resolution, a method needs to be developed to scale them to observations. Only then will we be able to apply them in a process-based approach to calculate realistic high-resolution $^{222}$Rn fluxes for Europe that compare well with observations, also in their absolute values. This task is part of the European EMPIR project

traceRadon (https://www.euramet.org/research-innovation/search-research-projects/details/project/radon-metrology-for-use-in-climate-change-observation-and-radiation-protection-at-the-environmental/), which will also conduct dedicated campaigns of quasi-continuous $^{222}$Rn flux and soil moisture measurements. With this objective, it has the potential to deliver a much more detailed data set to validate the new map and increase the observational basis also at ICOS stations to apply the Radon Tracer Method in the future.

**5   Conclusions**

The Radon Tracer Method provides a useful observation-based top-down tool to evaluate bottom-up inventories of greenhouse and other trace gas fluxes with a homogeneous source distribution similar to that of $^{222}$Rn. Applying the RTM for quantitative flux estimation relies on the accuracy of the $^{222}$Rn flux in the catchment area of the station. Its application for CH$_4$ at the Heidelberg measurement station had serious limitations due to the large heterogeneity of emissions in the

catchment area, which caused a huge variability of CH$_4$/$^{222}$Rn ratios. Large point source emissions were not captured by the RTM, thus under-estimating the total flux. Results of GHG flux estimates further depend on the parameters used to apply the RTM, such as the night-time period chosen as well as the requested quality of the regression (R$^2$). Only slightly changing these parameters, e.g. extending or reducing the night-time regression period by 2 hours or choosing an R$^2$ cut-off value of 0.8 rather than 0.7 introduces systematic differences of several percent each. Quantitative comparison of RTM-based with

bottom-up emission data is not directly possible without reliable footprint modelling of the nighttime observations. This may be hampered by the reliability of nighttime model transport; but applying the RTM also on model results may be an





appropriate way to circumvent this deficit. The model resolution should, however, be good enough to realistically represent the real source heterogeneity in the footprint of the station, in particular concerning point source emissions, so that model results are comparable with the observations. The caveat will then be that also the model-based RTM estimates will be

biased low. Therefore, in order to make reliable quantitative trace gas flux estimates with the RTM the unknown trace gas emissions should be distributed as homogeneously as possible. In Heidelberg, the top-down estimated $CH_4$ trend showing a 30% reduction of emissions from the mid-1990s to the mid-2000s compared well with the bottom-up EDGARv6.0 emission trend. But we could not observe a significant decrease of emissions thereafter, a sign that further efforts to reduce $CH_4$ emissions have not yet been successful in our Heidelberg catchment area.

**Appendix A**

In order to estimate the potential day-to-day variability of the $^{222}Rn$ flux from a typical soil in the Heidelberg catchment, we use the daily mean measurements of soil moisture (Fig. A1 upper panel) and temperature in the upper 30 cm of the Grenzhof soil (Wollschläger et al., 2009). We estimate the $^{222}Rn$ flux j for this site close to Heidelberg according to Karstens et al. (2015, their Eq. 8):

$$j(z=0) = -Q\sqrt{\frac{D_e}{\lambda}}$$

(A1).

We use a $^{222}Rn$ source strength of the soil material of $Q = 40$ mBq m$^{-3}$ s$^{-1}$, chosen such that the mean $^{222}Rn$ flux for 2007 and 2008 fits the average extrapolated flux for our small catchment area of 18.3 mBq m$^{-2}$ s$^{-1}$. $\lambda$ is the decay constant of $^{222}Rn$ (2.0974 10$^{-6}$ s$^{-1}$). The effective diffusivity $D_e$ is calculated according to Millington and Quirk (1960) from the molecular diffusivity of $^{222}Rn$ in air ($D_{a0} = 1.1 \cdot 10^{-5}$ m$^2$ s$^{-1}$), the measured total porosity of the Grenzhof soil ($\theta_p = 0.395$, Schmitt et al.,

2009) and the measured water-filled porosity $\theta_w$ (with $\theta_a = \theta_p - \theta_w$)

$$D_{e0} = D_{a0}\frac{\theta_a^2}{\theta_p^{\frac{2}{3}}} = D_{a0}\frac{(\theta_p - \theta_w)^2}{\theta_p^{\frac{2}{3}}}$$

(A2).

The dependency of the effective diffusivity on temperature was calculated according to Schery and Wasiolek (1998)

$$D_e(T) = D_{e0}\left(\frac{T}{273\,K}\right)^{\frac{3}{2}}$$

. (A3)

The day-to-day $^{222}Rn$ flux variability for 2007-2008 is displayed in the lower panel of Fig. A1.



**Code and data availability**

CH₄ and ²²²Rn data as well as computational codes will be made available at the ICOS carbon Portal ([https://www.icos-cp.eu/](https://www.icos-cp.eu/)).

**Author contributions**

IL designed the study together with UK and SH. IL evaluated the data and wrote the manuscript with help of all co-authors. SH was responsible for the CH₄ measurements. JD and MG conducted ²²²Rn observations and evaluated the data. UK contributed STILT footprint and concentration modelling and, together with FM programmed the evaluation codes.

**Acknowledgements**

The long-term atmospheric observations of CH₄ and ²²²Radon in Heidelberg have been conducted in the framework of numerous research projects funded by German Ministries and by the European Commission. These measurements are now part of the observational program at the ICOS pilot station of the Central Radiocarbon Laboratory of ICOS RI. Ute Karstens is partly funded by the metrology project EMPIR 19ENV01 "traceRadon".

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



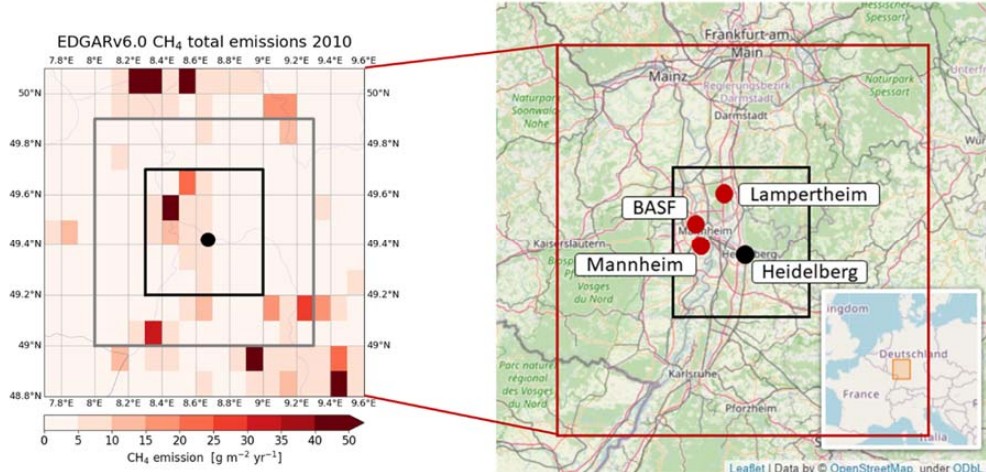


**Figure 1, right panel: Map of the upper Rhine valley south of Frankfurt/Main with the location of Heidelberg (black dot). The red dots indicate industrial areas (Mannheim/Ludwigshafen with the BASF chemical factory) as well as locations of large solid waste deposits (Lampertheim, Mannheim) in the small catchment of the station (© OpenStreetMap contributors 2021. Distributed under the Open Data Commons Open Database License (ODbL) v1.0, 2021). Left panel. Gridded CH4 emissions as reported by**

**the EDGARv6.0 inventory for 2010 (Crippa et al., 2021) covering a ca. 150 km x 150 km ("large") area surrounding Heidelberg. Two smaller areas, the so-called "small" (ca. 70 km x 70 km) and "intermediate" (ca. 110 km x 110 km) catchment areas of Heidelberg are marked as black and grey rectangle, respectively. Long-term trends of average CH₄ emissions from the three catchment areas are displayed in Fig. 3.**






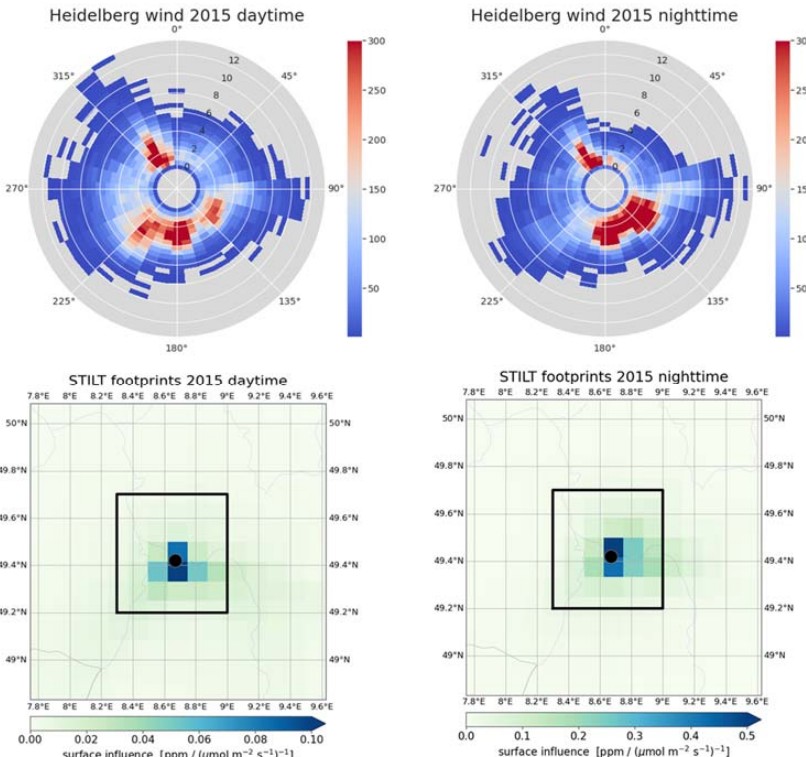

Figure 2: The upper two panels show the wind distributions (5-minute mean values, wind velocity in m s$^{-1}$ displayed on the radius) in 2015 measured on the roof of the Institute for Environmental Physics building at a height of 37 m a.g.l. Daytime (left panel) and nighttime (right panel) wind distributions are similar. The lower two panels show the annually aggregated surface influences of potential emissions for 2015 (left: daytime and right: nighttime). Note the different scales for day and night, indicating an appr. 5-fold sensitivity of emissions on concentrations observed at 30 m a.g.l. during nighttime compared to daytime. The black rectangle marks the "small" catchment area with Heidelberg in the approximate centre (black dot).






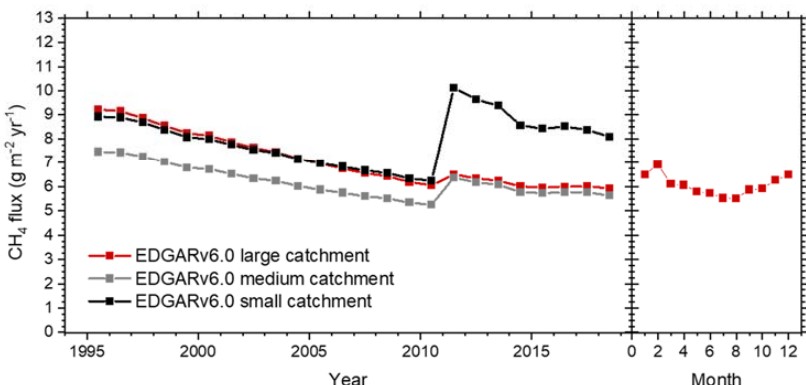


**Figure 3: Long-term trends of CH₄ fluxes as reported by the EDGARv6.0 emission inventory (Crippa et al., 2021). Trends for all three catchment areas show a significant decrease from 1995 to 2010 of about 30%. In 2011 an abrupt increase is observed, which is largest for the small catchment and due to an artefact of reported emissions in the MA/LU pixel (see text). The seasonal cycle of**
**2010 emissions in the large catchment is displayed on the right hand side of the diagram.**

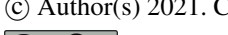



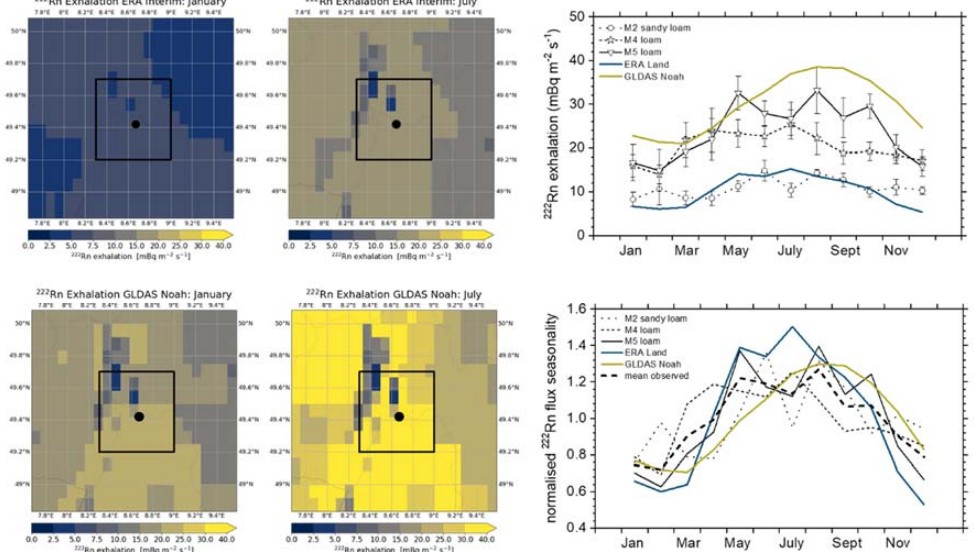

Figure 4, left panels: $^{222}$Rn exhalation rates as estimated by Karstens et al. (2015) for the large Heidelberg catchment area based on the ERA Interim Land (upper panels) and GLDAS Noah (lower panels) soil moisture models for January (left) and July (middle). The small catchment area is marked by the black rectangle with Heidelberg in its appr. centre (black dot). The very low $^{222}$Rn fluxes north-west of Heidelberg stem from the $^{222}$Rn flux limitation assumed in Karstens et al. (2015) based on the water table depth map by Miguez-Macho et al. (2008). The upper right panel shows the mean seasonal cycle of the modelled fluxes in comparison to measurements conducted south of Heidelberg on sandy loam (M2) and loamy soils (M4, M5). Normalised (to their annual means) seasonal cycles of the fluxes shown in the upper right panel are displayed in the lower right panel. The mean observed $^{222}$Rn flux seasonality is also shown as thick dashed line.





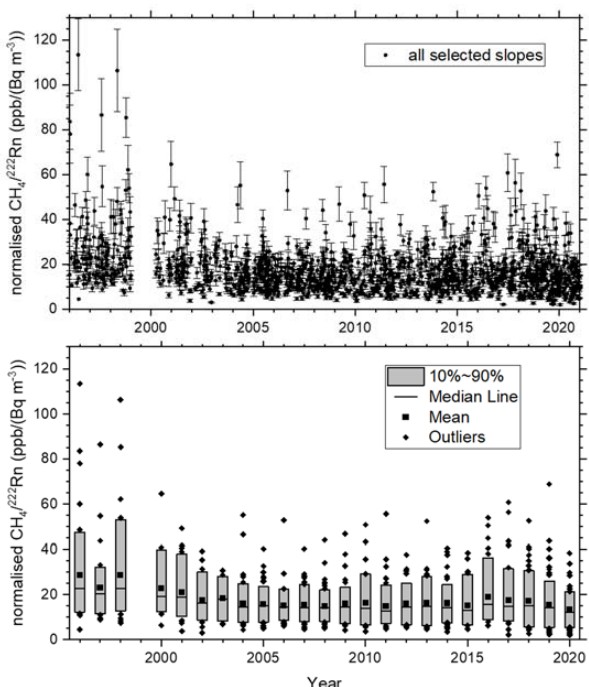

**Figure 5, upper panel: Individual normalised $CH_4/^{222}Rn$ slopes and their 1σ uncertainties of linear regressions with $R^2 \geq 0.7$,**
**calculated from half-hourly night time (21:00 h to 04:00 h CET) data. Lower panel: annually aggregated $CH_4/^{222}Rn$ slopes**
**presented as box-plots with the boxes including 80% of the data.**





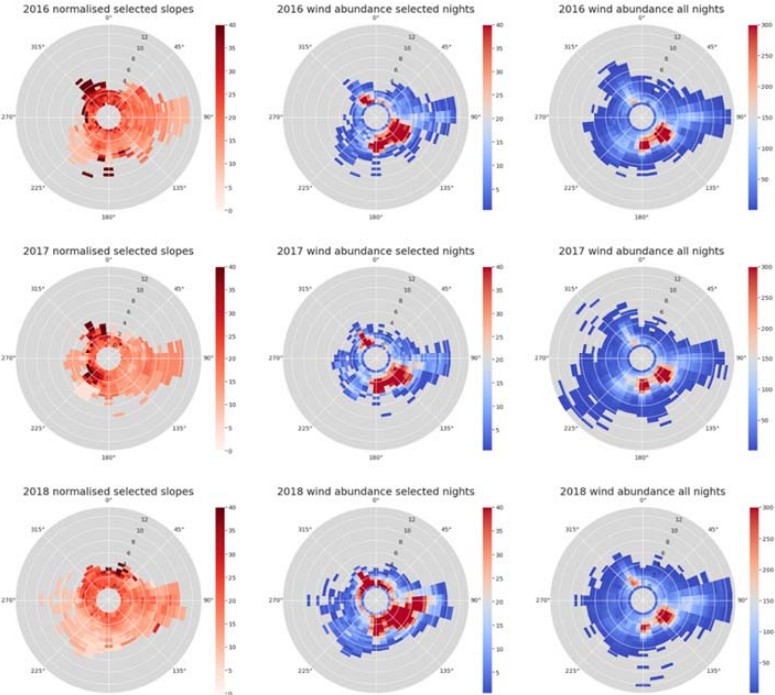

**Figure 6 left column: Distribution of night time slopes (in ppb (Bq m⁻³)⁻¹) by local wind direction (°) and velocity (m s⁻¹) for the years 2016, 2017 and 2019. The corresponding frequency distributions of wind direction and velocity for the selected nights are displayed in the second column while the distribution for all nights of the respective year (from 21:00 h – 04:00 h CET) are shown in the third column. It is clearly visible that wind velocities are generally lower during the selected nights than during all nights.**






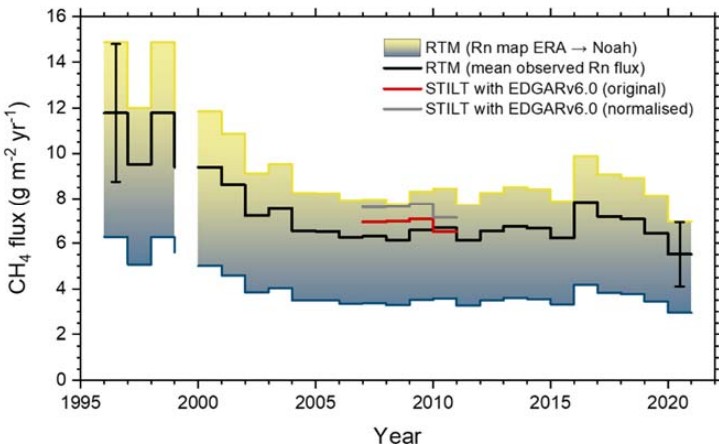

Figure 7: Long-term trend of the RTM-based CH₄ flux in the Heidelberg catchment area. The black histogram (with typical RTM-based uncertainties shown for the first and the last year of observations) was calculated based on the observation-based $^{222}$Rn flux of 18.3±4.7 Bq m$^{-2}$ s$^{-1}$. The coloured area shows the range of RTM-based CH₄ flux estimates if either the GLDAS Noah soil moisture (yellow) or the ERA Interim Land soil moisture (blue) based $^{222}$Rn flux average of the small catchment area would have

been used to calculate RTM-based CH₄ fluxes. Also included in the diagram are RTM-based results from STILT-modelled CH₄ and $^{222}$Rn data for 2007 – 2010 (based on the slopes in Fig. 8). The red line shows the original results using the EDGARv6.0 emission inventory and the $^{222}$Rn flux climatology while the grey line shows the STILT results normalised to the observation-based $^{222}$Rn flux (see text).





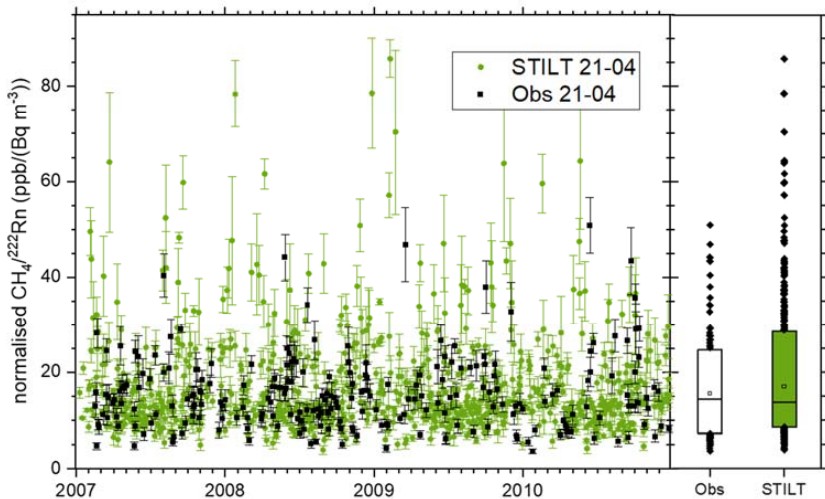


**Figure 8: Variability of observed (Obs, black squares) and simulated (STILT, green dots) night-time $CH_4/^{222}Rn$ slopes from 2007 to 2010 (left panel). The right panel shows the distributions of all slopes with the boxes including 80% of the data, the open squares representing the mean and the horizontal lines the median values. Note that for the further discussion we excluded the three modelled values >70 ppb (Bq m$^{-3}$)$^{-1}$.**


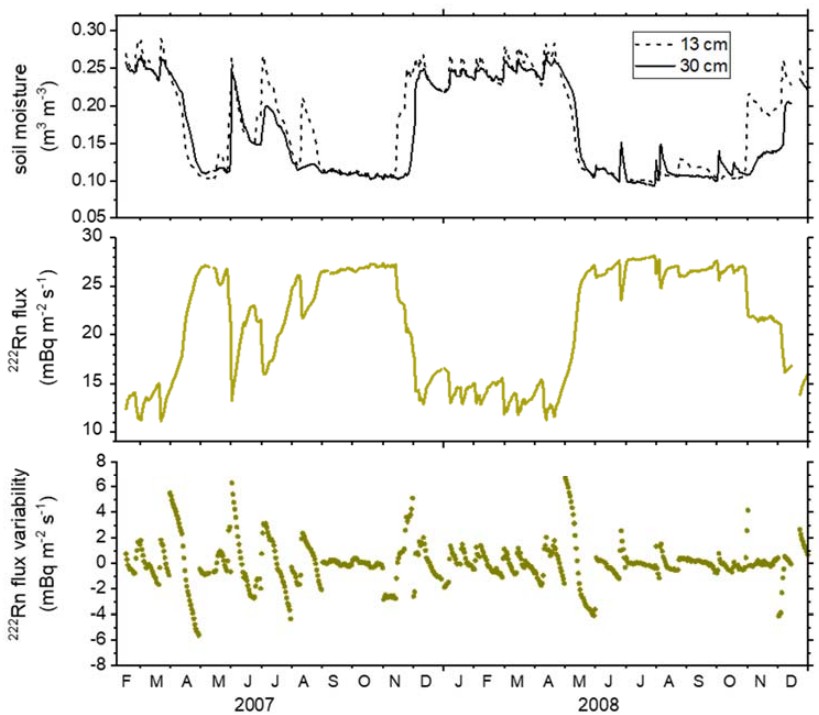

**Figure A1, upper panel: Daily variations of measured soil moisture at the Grenzhof site near Heidelberg at 13 cm and 30 cm depth. The hypothetical $^{222}$Rn flux estimated from the soil moisture (and temperature) variability is shown in the middle panel, while the day-to-day variability around the corresponding monthly means of the $^{222}$Rn flux is shown in the lowest panel. The average variability corresponds to 10% around the monthly mean flux.**