# Peer review of "Limitations of the Radon Tracer Method (RTM) to estimate regional Greenhouse Gases (GHG) emissions – a case study for methane in Heidelberg"

_Atmospheric Chemistry and Physics, 2021_

## Referee Comment (RC1)

**Review**

**Atmospheric Chemistry and Physics* Manuscript #2021-661**

Limitations of the Radon Tracer Method (RTM) to estimate regional Greenhouse Gases (GHG) emissions – a case study for methane in Heidelberg

(Levin, Karstens, Hammer, DellaColetta, Maier and Gachkivskyi)

23rd September 2021

Overall recommendation: Accept after minor changes

**General comments**

This is a very interesting and valuable study applying the Radon Tracer Method (RTM) to estimate trends in nocturnal methane emissions in a complex region around Heidelberg over the period 1996-2020 and comparing the results to EDGARv6.0 bottom-up inventories. Emphasis is placed on the shortcomings of the technique, particularly in the critical importance of having accurate and representative knowledge of radon emissions from soils in the flux footprint, including the influence of time-varying soil moisture, and the interference of significant point-source pollution in the calculations.

This paper contributes topical original research falling within the scope of *Atmospheric Chemistry and Physics*. The manuscript is methodical, clearly written and logically structured. The experimental design is appropriate, and the authors utilise a range of appropriate analysis techniques and visual presentation tools to illustrate the relevant information required to support their arguments and conclusions. The outcomes and implications of the study are well summarised in the Conclusions and Abstract.

I am very happy with the paper and recommend acceptance after attention has been paid to a few minor issues listed below.

**Specific comments**

(1)  I believe this manuscript would benefit greatly by clearer elucidation of the role of the "nocturnal accumulation" RTM within the broader context of European top-down trace gas emission estimates using radon. This could be accomplished quite easily by adding short paragraphs in the Introduction, Methods and Discussion sections (suggested locations are provided below under "**Minor and technical comments /**

**suggestions**"), outlining the differences in scope and implementation between the "nocturnal accumulation" (this paper), "tall tower" and "baseline" (mountaintop and remote location) applications of RTM, and emphasizing their complementarity. The "nocturnal accumulation" RTM, applied in the current study, uses surface-based measurements for estimating local fluxes (say, up to 200km spatial scale), and should be contrasted with the RTM as applied to measurements from tall towers, which estimates fluxes up to the regional scale (200-1000km). In the latter case, trace gases are monitored in the deeper mixed / residual layer above the nocturnal inversion and are therefore integrative of the whole boundary layer, the entire diurnal cycle and much bigger fetch areas (regional to continental scale). For these reasons, they are not restricted to nocturnal-only measurements and do not suffer so much from the problem of representing local point sources within the footprint (the strong boundary layer regional mixing process tends to increase the comparability of the trace gas and radon signals). However, they require different assumptions about reference ("background") signals and exchanges with the free troposphere, and have their own special implementation difficulties (e.g., increased uncertainty in the definition of the footprint, losses/gains at the boundaries and the top of the box, non-stationarity due to synoptic weather influences etc.). Finally, RTM applications at baseline stations (mountaintops and remote locations) are similar in implementation to the tall tower case and can be used to estimate fluxes from regional to continental and even hemispheric scales.

(2) With regards to the discussions on the effects of point source emissions on the RTM results: If point source emissions are injected directly into the nocturnal inversion layer, or if they are injected above (i.e., from tall stacks) but are then fully or partially incorporated into the inversion layer by subsequent "fumigation" events, then they may be mixed in the footprint of the measurement site and influence the average trace gas levels experienced on a given night. If this is an uncommon occurrence, it will be dismissed as an outlier in the analysis. However, if it happens often, then it may end up being correlated with the radon observations because both scalars are mixed (or partially mixed!) within the same nocturnal volume. In other words, this could lead to a range of scatt in the correlation plots...

(3) Seasonal variations in the radon flux translate to seasonal variations in the measured atmospheric radon concentrations. The latter are only partially matched by corresponding variations in the measured $CH_4$ concentrations, resulting in a residual seasonality in the computed $CH_4/^{222}Rn$ ratios. This latter seasonality is initially removed from the ratios, so that a focus can be placed on the effects of the absolute flux errors. The intention appears to be (according to the first paragraph of Section 3.1) that the seasonality in the ratios would be returned to later for separate investigation; however, this is never done.

(4) There is no comment anywhere (unless I missed it?) on the bias introduced into the trace gas flux estimations by the fact that only nocturnal measurements are used in this flavour of the RTM. If there is a strong diurnal variation in the fluxes estimated by the nocturnal RTM method, then the results will not be an accurate representation of the diurnal average flux (e.g., $CO_2$ will only deliver respiration fluxes). This should perhaps be noted in the description of the method, along with a justification for why the problem "might not be too bad" for $CH_4$.

Minor and technical comments / suggestions

(1) [Abstract] p2, line 16: Here and elsewhere in the manuscript, the authors might consider changing "catchment area" to "flux footprint" or similar. In my experience, the word "catchment" is a hydrological term that refers specifically to an area defined by a watershed (topographical high-altitude line).

(2) [Abstract] p2, line 18: Change "total $CH_4$ emission" to "total nocturnal $CH_4$ emission".

(3) [Abstract] p2, line 19: Change "exhalation rate from soils" to "exhalation rates estimated from soils".

(4) [Abstract] p2, line 23: Change "RTM-based top-down with bottom-up" to "RTM-based top-down flux estimates with bottom-up".

(5) [Abstract] p2, line 26: Change "as their emissions are not captured by the RTM method" to "as their emissions may not be fully captured by the RTM method, for example if stack emissions are injected above the top of the nocturnal inversion layer, or if point source emissions from the surface are not well mixed into the footprint of the measurement site".

(6) [Introduction] p2, line 37: Change "(UNFCCC, 2015). But only the" to "(UNFCCC, 2015), but only the" OR "(UNFCCC, 2015). However, the".

(7) [Introduction] p2, line 41: Change "A possibility to estimate continental" to "A possibility for estimating continental nocturnal".

(8) [Introduction] p3, line 45: Change "lifetime of about 5.5 days" to "half-lifetime of about 3.8 days".

(9) [Introduction] p3, line 48: After "Liu et al., 1984", you might consider adding "Williams et al., 2011". Reference: Williams AG, Zahorowski W, Chambers SD and Griffiths A, 2011: The vertical distribution of radon in clear and cloudy daytime terrestrial boundary layers. *J Atmos Sci.* **68**:155–174.

(10) [Introduction] p3, line 50: Change "correlated increases" to "correlated overnight increases".

(11) [Introduction] p3, line 52: Change "recommended to use this tracer for transport model validation but also to apply the RTM" to "recommended for use in transport model validation and application of the RTM".

(12) [Introduction] p3, end of line 53: This might be a good place to remind the reader that the "nocturnal accumulation" application of the RTM is significantly different from "tall tower" RTM applications. See "**Specific comments**" #1.

(13) [Introduction] p3, line 65: Change "when missing precipitation" to "when a lack of precipitation".

(14) [Introduction] p3, end of line 68: It would be helpful here to have a short summary of the known effects of increasing near-surface soil moisture on the radon flux. For example, is it a linear / logarithmic relationship, or is it a negligible effect until the soil gets very close to saturation? This would help the reader to get a feel for the potential severity of the problem and prepare them for your discussion of radon fluxes around Heidelberg in later sections.

(15) [Introduction] p3, line 73: Remove "and $CH_4$". Otherwise, it is a circular statement ("we use $CH_4$ flux variability to evaluate $CH_4$ emission estimates").

(16) [Methods 2.1] p4, lines 89-91: Consider enhancing the discussion of H(t) like this: H(t) is a length scale corresponding to the 'effective' depth that the stable layer would have if the trace gases of interest were uniformly mixed vertically within it. The layer is

assumed to be mixed 'well enough' that the measured near-surface concentrations can be considered as representative of the layer-averaged values (Williams et al., 2016). Reference: Williams AG, Chambers SD, Conen F, Reimann S, Hill M, Griffiths AD and Crawford J, 2016: Radon as a tracer of atmospheric influences on traffic-related air pollution in a small inland city. *Tellus B* **68**, 30967. http://dx.doi.org/10.3402/tellusb.v68.30967

(17) [Methods 2.1] p5, lines 107-108: "… residual layer air that has a $CH_4/^{222}Rn$ ratio similar to that at the start of the night-time observation period". I assume this is the value you use to define the reference point for $\Delta C$ in the equations above? If so, then maybe mention that here. The encroachment of residual layer air into the growing nocturnal boundary layer is also discussed by Williams et al. (2016): see ref above.

(18) [Methods 2.1] p5, line 113: After "and the trace gas", consider adding ", or at least that surface source functions can be considered to be essentially random and uncorrelated with atmospheric processes operating on short temporal and small spatial scales".

(19) [Methods 2.1] p5, lines 113-120: With regards to the discussions on the effects of point source emissions on the RTM results, you could discuss this further as per "**Specific comments**" #2.

(20) [Methods 2.1] p5, line 116: Change "relevant" to "present".

(21) [Methods 2.2] p6, line 168: "… this method is only applicable for area sources that are similarly homogeneously distributed as those of $^{222}Rn$ (Eq. 4)". This is true, but see "**Specific comments**" #2.

(22) [Methods 2.2] p7, line 169: Maybe change "be missing" to "be wholly or partially missing". See "**Specific comments**" #2.

(23) [Methods 2.2] p7, lines 174-176: "In the inventories these fluxes are associated with much larger uncertainties than those from point sources and are thus a rewarding target for the RTM". This is an excellent point! See "**Specific comments**" #1.

(24) [Methods 2.3] p7, line 179: "The most important pre-requisite to apply the Radon Tracer Method for quantitative GHGs flux estimates are representative $^{222}Rn$ soil exhalation rates in the catchment area". Maybe you should remind the reader here that Eqn (4) implies that errors in the derived GHG fluxes will be underline{directly proportional} to errors in the radon fluxes used.

(25) [Methods 2.3] p7, line 198: Change "from the sandy soils of M1 and M3" to "from sandy soils (denoted M1 and M3)".

(26) [Methods 2.3] p8, line 206: Change "from M2, M4 and M5" to "from soil types denoted M2, M4 and M5".

(27) [Methods 2.3] p8, lines 216-218: Change "This seasonality… lower right panel of Fig. 4" to "This seasonality in the radon fluxes leads to a seasonal variation in atmospheric radon concentrations which then transfers to the computed $CH_4/^{222}Rn$ ratios because the corresponding $CH_4$ seasonality is relatively small in amplitude. In order to investigate this seasonality separately from the overall effects of the absolute flux errors, the measured and modelled seasonality of $^{222}Rn$ fluxes in the two pixels south of Heidelberg were first normalised to the respective annual means and are shown in the lower right panel of Fig. 4. This will be discussed further in the Results section".

(28) [Results 3.1] p10, line 291: After "during all nights", maybe add a description of typical conditions during excluded nights. For example: "Nights excluded by this restriction tended to have higher wind speeds, be cloudy or were disturbed by passing synoptic weather patterns (e.g., fronts)".

(29) [Results 3.1] p10, line 295: Change "the $^{222}$Rn exhalation rate from soils has a pronounced seasonality" to "as discussed in Section 2.3, the measured and modelled $^{222}$Rn exhalation rates from soils both exhibit a pronounced seasonality".

(30) [Results 3.1] p10, lines 297-298: Change "This seasonality of the $^{222}$Rn flux imposes a seasonality on the $CH_4/^{222}$Rn ratios. We therefore normalised…" to "This seasonality of the $^{222}$Rn flux results in a seasonality in atmospheric radon concentrations and consequently also the computed $CH_4/^{222}$Rn ratios (as the corresponding $CH_4$ seasonality is relatively small in amplitude). In the analysis to follow, we first normalised...".

(31) [Results 3.1] p11, lines 297-299: Change "to the annual mean $^{222}$Rn flux" to "that adjusts the $^{222}$Rn flux to its annual mean value".

(32) [Results 3.1] p11, lines 300-301: Change "This intermediate step was taken because of the large uncertainty of the *absolute* $^{222}$Rn flux in contrast to its much better defined seasonality" to "This intermediate step was taken in order to separately study both the large uncertainty of the *absolute* $^{222}$Rn flux and its much better defined seasonality". See "**Specific comments**" #3.

(33) [Results 3.1] p11, lines 325-328: "As mentioned … afternoon before". It would be really nice to see an illustration of this by showing examples of $^{222}$Rn and $CH_4$ hourly time series for two contrasting nights characterized by strong positive and strong negative correlations. In the latter case, is the computed equivalent mixing layer depth H close to 30m?

(34) [Results 3.2] p12, lines 334-336: Change "The $CH_4/^{222}$Rn slopes displayed … in the footprint of Heidelberg" to "The $CH_4/^{222}$Rn slopes displayed in Fig. 5 show large variability. It is of interest to explore if this variability can be explained by spatial variations in the $CH_4$ emissions, and if so, the extent to which we can associate the high-slope cases with hot spot emission areas in the footprint of Heidelberg".

(35) [Results 3.2] p12, line 336: Change "air mass influence" to "air mass footprint".

(36) [Results 3.2] p12, line 338: Change "origin is from" to "has passed over".

(37) [Results 3.2] p12, line 358: Change "will not be captured" to "may not be fully captured". See my previous comments.

(38) [Results 3.2] p12, line 360: Change "can we" to "we can".

(39) [Results 3.4] p13, line 384: Change "M2, M4 and M5 to" to "M2, M4 and M5 to be".

(40) [Results 3.4] p13, line 385: Change "The corresponding $CH_4$ flux it is plotted as" to "The corresponding calculated $CH_4$ flux is plotted as the".

(41) [Discussion 4.1] p16, line 483: Change "captured" to "fully captured".

(42) [Discussion 4.2] p18, line 543-544: Change "could show" to "have shown".

(43) [Discussion 4.2] p18, line 545: Change "ask for" to "dictate a need for".

(44) [Discussion 4.2] p18, lines 546-550: "A second problem … less well-defined $^{222}$Rn fluxes". I think a slightly more detailed discussion is needed here. See my suggestions in "**Specific comments**" #1.

(45) [Conclusions 5] p19, line 583: Change "quantitative flux estimation relies on the accuracy" to "quantitative flux estimation relies critically on the accuracy".

(46) [Conclusions 5] p19, line 583: Change "catchment" to "footprint".

(47) [Conclusions 5] p19, line 585: Change "catchment" to "footprint".

---

## Referee Comment (RC2)

**Review**

Atmospheric Chemistry and Physics Manuscript #2021-661

Limitations of the Radon Tracer Method (RTM) to estimate regional Greenhouse

Gases (GHG) emissions – a case study for methane in Heidelberg

(Levin, Karstens, Hammer, DellaColetta, Maier and Gachkivskyi)

27th September 2021

Overall recommendation: Accept after minor revisions

**General comments**

In this study the authors apply the Radon Tracer Method (RTM) using a 24-year dataset of atmospheric methane ($CH_4$) and radon ($^{222}Rn$) concentrations, measured at the Heidelberg city, to estimate the trend of methane emissions over the city surrounding area. Then they compare the RTM based $CH_4$ emissions with results obtained using the EDGARv6.0 bottom-up inventory. Authors, who were the first to introduce the RTM in Levin et al., 1999, also analyze the strength and weakness of the RTM application mainly in regards to the radon flux value used in it and the representability of the catchment area of the atmospheric station depending on the heterogeneity of the GHGs sources.

The aim of this paper follows completely within the scope of this journal. The study is well designed and the English of the manuscript has been already reviewed by the other reviewer, who is a native speaker. The work behind the achievement of the 24-year dataset is impressive and it gives a really robust statistics to the results obtained in this study.

On the other hand, some aspects of the manuscript could be improved for the fluency of the reading and to clearly identify what has been done so far in the field of the RTM application. The state of the art and the discussion of the results of this study are not updated and they did not consider past outcomes from others researchers.

The paper deserves to be published in the ACP journal after that some changes will be made as explained in details in the following sections.

**Specific comments**

**Section: 1. Introduction**

- In Lines 55-56 authors declare that RTM has been applied assuming a spatially homogenous radon flux (bibliography here stops to 2009 and they could also add Vogel et al., 2012 Wada et al., 2013 and Grossi et al., 2014). Furthermore, this sentence is not fully correct because Grossi et al., 2018 applied the RTM calculating the effective radon flux. This was calculated by coupling radon flux data, obtained using the output for the 40-year climatology obtained with the model developed by López-Coto et al. (2013), with the footprints calculated by the ECMWF-FLEXPART model (version 9.02) (Stohl, 1998) (For more info please look at the Figure 8 of Grossi et al., 2018 and equation n. 3). In addition, in Grossi et al., 2018 the $CH_4$ fluxes, obtained by RTM, were also compared for the first time with $CH_4$ fluxes obtained coupling the EDGAR inventory with ECMWF-FLEXPART footprints for the same period.

- In Lines 66-67 authors say that the basic assumption for the classical RTM application is of having a more or less constant radon flux. I do not personally agree with this and I think it already stays in the past. Nowadays it is known that for correctly applying the RTM we need high quality and reliable atmospheric GHGs and radon concentration data and validated radon flux models with as high as possible spatial and temporal resolution. These are actually between the main goals of the project EMPIR traceRadon (presented in Röttger et al., 2021) which wants to offer also a metrology for radon flux measurements and sensitivity studies for the RTM applications.

I think it will be nice to have all this previous information in the state of the art.

**Section: 2. Methods**

*Radon flux estimation for the RTM application*: The structure of this section does not help the reader to understand the methodology applied for the calculation of the radon flux used within the RTM. I had the impression that authors finally used a constant value of $18.3 \pm 4.7$ mBq m$^{-2}$ s$^{-1}$. Is it correct? Did you not estimated the effective flux seen by the station coupling radon flux climatology output from Karstens et al., 2015 with STILT footprints? It may help to have this info in a dedicated paragraph where the estimation of the radon flux used for the RTM is clearly explained.

*STILT footprints*: It will help to have a dedicated section where the calculation of the STILT footprint is described. How long were the back trajectories used for it? Which was the height of the boundary layer used in the STILT simulations for it? I was not able to find this info in the manuscript and it could be useful, as explained in Grossi et al., 2018, when effective radon flux is estimated using also model footprints and for future RTM applications protocols.

*$CH_4$ and $^{222}Rn$ measurements*: I agree with the authors on the importance of correctly estimating the radon flux values used in the RTM but equation 3 gives the same weight to methane and radon concentration measurements too. I think it will be nice to have a paragraph dedicated to experimental measurements ($CH_4$, $^{222}Rn$ and meteorology). Here, a 24-year dataset of radon progeny ($^{214}Po$) concentration measured using a static filter method (Levin et al., 2002), and a

constant disequilibrium correction factor between $^{222}$Rn and $^{214}$Po of 1.11 ($F_e$), has been used as explained in lines 275-282. However, results from inter-comparison studies between radon and radon progeny monitors based on different measurements techniques (Grossi et al., 2016; Schmithüsen et al., 2017; Grossi et al. 2020) show that under saturated atmospheric water conditions and low atmospheric aerosol concentration this disequilibrium factor $F_e$ could change inducing an underestimation of atmospheric radon activity concentration by static filter methods. In addition, Levin et al., 2017 estimated a correction factor to take into account the $^{222}$Rn progeny loss in long tubing based on static filter measurements in the laboratory and in the field. Taking into account all these previous outcomes, I wonder if authors have filtered they data for rain/low aerosol episodes and/or applied these corrections factors before using the dataset for RTM calculation. It should be clearly stated in the manuscript. Finally, authors say (Line 279) that they used an ANSTO scale to calibrate their instrument. Unfortunately ANSTO instruments, running at several ICOS stations, are calibrated using their own source which is located within each instrument. This means that there is not a primary standard or a second transfer standard instrument to harmonize these instruments and using a ANSTO scale does sound correct to be used. The lack of a ambient radon measurement metrology was one of the main aims of the traceRadon project. For example, in Grossi et al., 2020 the correlation of the same ARMON (Grossi et al., 2012) with two ANSTO monitors (located respectively at Saclay (100 m.a.g.l.) and at Orme de Marisier (5 m.a.g.l.)) had slopes of 0.97 ± 0.01 and 0.96 ± 0.01 with intercepts of 0.01 ± 0.06 and 0.01 ± 0.06, respectively. Schmithüsen et al., 2017 found a correlation, at the Heidelberg station, between the HRM and the ANSTO instrument of 1.22 ± 0.01 with an intercept of 0.42 ± 0.04.

**Section 3: Results**

*Paragraph 3.5*: The comparison of STILT based and RTM based results of the $CH_4$/$^{222}$Rn slopes is obtained, if I understood correctly, comparing the ratios between $CH_4$ and $^{222}$Rn concentrations simulated using the STILT model in forward mode and using, as emissions, the radon flux climatology output from the model presented in Karstens et al., 2015 with the ratios of measured $CH_4$/$^{222}$Rn. Is it correct? Authors said that these comparison seems to work properly. May this due to the fact that RTM is used only applying a constant radon flux values over the time and area where here the forward simulation is run with spatial radon flux climatology?

*Discussion and Conclusion*: Authors may revisit the discussion and conclusions sections taking into account the previous comments expressed by the reviewer.

**Minor and technical comments / suggestions**

- Please use radon or $^{222}$Rn instead of $^{222}$Radon within the manuscript. The same nomenclature for $^{226}$Radium and $^{214}$Polonium.
- Line 384 mBq instead of Bq.

**References to be included**

Grossi, C., Arnold, D., Adame, A. J., Lopez-Coto, I., Bolivar, J. P., de la Morena, B. A., and Vargas, A.: Atmospheric 222Rn concentration and source term at El Arenosillo 100 m meteorological tower in southwest Spain, Radiat. Meas., 47, 149–162, https://doi.org/10.1016/j.radmeas.2011.11.006, 2012.

Grossi, C., Àgueda, A., Vogel, F. R., Vargas, A., Zimnoch, M., Wach, P., Martín, J. E., López-Coto, I., Bolívar, J. P., Morguí, J.-A., and Rodó, X.: Analysis of ground-based 222Rn measurements over Spain: filling the gap in southwestern Europe, J. Geophys. Res.-Atmos., 121, 11021–11037, https://doi.org/10.1002/2016JD025196, 2016.

Grossi, C., Vogel, F. R., Curcoll, R., Àgueda, A., Vargas, A., Rodó, X., and Morguí, J.-A.: Study of the daily and seasonal atmospheric CH4 mixing ratio variability in a rural Spanish region using 222Rn tracer, Atmos. Chem. Phys., 18, 5847–5860, https://doi.org/10.5194/acp-18-5847-2018, 2018

Grossi, C., Chambers, S. D., Llido, O., Vogel, F. R., Kazan, V., Capuana, A., Werczynski, S., Curcoll, R., Delmotte, M., Vargas, A., Morguí, J.-A., Levin, I., and Ramonet, M.: Intercomparison study of atmospheric 222Rn and 222Rn progeny monitors, Atmos. Meas. Tech., 13, 2241–2255, https://doi.org/10.5194/amt-13-2241-2020, 2020.

Röttger et al 2021 Meas. Sci. Technol. in press https://doi.org/10.1088/1361-6501/ac298d

Vogel, F. R., Ishizawa, M., Chan, E., Chan, D., Hammer, S., Levin, I., and Worthy, D. E. J.: Regional non-CO2 greenhouse gas fluxes inferred from atmospheric measurements in Ontario, Canada, J. Integr. Environ. Sci., 9, 45–55, https://doi.org/10.1080/1943815X.2012.691884, 2012.

Wada, A., Matsueda, H., Murayama, S., Taguchi, S., Hirao, S., Yamazawa, H., Moriizumi, J., Tsuboi, K., Niwa, Y., and Sawa, Y.: Quantification of emission estimates of CO2, CH4 and CO for East Asia derived from atmospheric radon-222 measurements over the western North Pacific, Tellus B, 65, 18037, https://doi.org/10.3402/tellusb.v65i0.18037, 2013.

---

## Author Comment (AC1)

Reply to specific comments #1 - #4 of Reviewer 1, Alastair Williams

We wish to thank Alastair Williams for his insightful and timely review that gives us the opportunity for further discussion before revising our manuscript (our replies are printed in blue).

(1) I believe this manuscript would benefit greatly by clearer elucidation of the role of the "nocturnal accumulation" RTM within the broader context of European top-down trace gas emission estimates using radon. This could be accomplished quite easily by adding short paragraphs in the Introduction, Methods and Discussion sections (suggested locations are provided below under "**Minor and technical comments / suggestions**"), outlining the differences in scope and implementation between the "nocturnal accumulation" (this paper), "tall tower" and "baseline" (mountaintop and remote location) applications of RTM, and emphasizing their complementarity. The "nocturnal accumulation" RTM, applied in the current study, uses surface-based measurements for estimating local fluxes (say, up to 200km spatial scale), and should be contrasted with the RTM as applied to measurements from tall towers, which estimates fluxes up to the regional scale (200-1000km). In the latter case, trace gases are monitored in the deeper mixed / residual layer above the nocturnal inversion and are therefore integrative of the whole boundary layer, the entire diurnal cycle and much bigger fetch areas (regional to continental scale). For these reasons, they are not restricted to nocturnal-only measurements and do not suffer so much from the problem of representing local point sources within the footprint (the strong boundary layer regional mixing process tends to increase the comparability of the trace gas and radon signals). However, they require different assumptions about reference ("background") signals and exchanges with the free troposphere, and have their own special implementation difficulties (e.g., increased uncertainty in the definition of the footprint, losses/gains at the boundaries and the top of the box, non-stationarity due to synoptic weather influences etc.). Finally, RTM applications at baseline stations (mountaintops and remote locations) are similar in implementation to the tall tower case and can be used to estimate fluxes from regional to continental and even hemispheric scales.

We fully agree with the reviewer that $^{222}$Radon in combination with greenhouse gases (GHG) observations has potential to improve top-down estimates of GHG emissions. A review paper on the various applications would certainly be a useful contribution to the scientific literature. It was, however, not our aim to provide such a review. Our focus was only on the "nocturnal accumulation" RTM, also because this is currently the most frequent application of the RTM that reports **quantitative** regional trace and GHGs emissions estimates.

Using larger scale representative $^{222}$Radon observations from coastal, tall tower or mountain stations e.g. for air mass characterisation (marine vs. continental) or for transport model validation (e.g. validating nocturnal boundary layer thickness) is certainly another valuable application of the "RTM". However, if such applications shall provide quantitative results, they will also be restricted by the not well known $^{222}$Radon flux from continental soils. In the regional Heidelberg study area, we have the advantage that soil types, their textures and Uranium contents are rather homogeneous. This allowed us to use our long-term $^{222}$Radon flux observations instead of flux model data to estimate RTM-based $CH_4$ emissions. However, a larger footprint (200-1000km) will cover soil types that are much more variable in their $^{222}$Radon flux than in the restricted upper Rhine valley. As the reviewer states correctly, this would probably increase the uncertainty of footprint-weighted RTM-based fluxes considerably. But, no matter what the scale is, local, regional, continental or even hemispheric, in all applications using $^{222}$Radon as a **quantitative** tracer, the big unknown will be the soil exhalation rate. To our knowledge, there currently exist four different $^{222}$Radon flux maps for Europe developed using different input data and parameterisations. Their long-term medium emission rates differ by a factor of two (see Karstens et al., 2015, Fig. 4). We, thus, only have to hope for a significant improvement of the reliability of soil moisture models, the most important but very variable input

parameter to accurately calculate $^{222}$Radon exhalation rates. Note, however, that long-term continuous soil moisture measurements are notoriously difficult to conduct, and we just recently learned that in many cases, they are not precise (besides being not representative) enough to validate modelled soil moistures. At larger scales and in a well-mixed atmospheric boundary layer, the spatial heterogeneity of the $^{222}$Radon, and also of the GHG flux may level out, but the absolute mean $^{222}$Radon flux from the different continents will stay as unknown with uncertainties of the same order or even larger than the uncertainty of bottom-up GHGs emission inventories (at least in Europe).

We admit that this is a not very promising message. Therefore, we hesitate to further promote the RTM and other $^{222}$Radon applications - for quantitative use - in this study. We rather feel that the Radon community has eventually to face that message and start focussing on extensive $^{222}$Radon flux measurement campaigns in the catchments of atmospheric monitoring stations as well as on developing better soil models (for moisture and for gas transport).

(2) With regards to the discussions on the effects of point source emissions on the RTM results: If point source emissions are injected directly into the nocturnal inversion layer, or if they are injected above (i.e., from tall stacks) but are then fully or partially incorporated into the inversion layer by subsequent "fumigation" events, then they may be mixed in the footprint of the measurement site and influence the average trace gas levels experienced on a given night. If this is an uncommon occurrence, it will be dismissed as an outlier in the analysis. However, if it happens often, then it may end up being correlated with the radon observations because both scalars are mixed (or partially mixed!) within the same nocturnal volume. In other words, this could lead to a range of scatt in the correlation plots...

We agree with the reviewer and confirm that there were indeed very few occasions, also in the observations, with very high $CH_4/^{222}Rn$ slopes, which met the selection criterion and showed stable trajectories with air mass origin from the largest $CH_4$ point sources located in the north-west of Heidelberg (MA/LU).

(3) Seasonal variations in the radon flux translate to seasonal variations in the measured atmospheric radon concentrations. The latter are only partially matched by corresponding variations in the measured $CH_4$ concentrations, resulting in a residual seasonality in the computed $CH_4/^{222}Rn$ ratios. This latter seasonality is initially removed from the ratios, so that a focus can be placed on the effects of the absolute flux errors. The intention appears to be (according to the first paragraph of Section 3.1) that the seasonality in the ratios would be returned to later for separate investigation; however, this is never done.

The reviewer is correct – we were so much overwhelmed by the large variability of the normalised $CH_4/^{222}Rn$ slopes, that we decided to only focus on annual mean slopes and corresponding RTM-based $CH_4$ fluxes and finally neglected analysing the results with respect to a possible seasonal variation of the RTM-based $CH_4$ emissions. We now made a respective analysis by calculating normalised monthly mean slopes for the years 2004-2015, i.e. the period when the estimated annual mean fluxes varied from year to year by only 3%. Monthly mean slopes for the individual years, normalised to the long-term mean of all slopes, are plotted in Fig. 1 (left panel), together with the mean seasonal cycle of EDAGARv6.0 $CH_4$ emissions in 2010 from the large catchment area (right panel of Fig. 3 of the manuscript). The right panel of Fig. 1 shows the mean seasonal cycle and its standard deviation for 2004-2015. The error bars also include the uncertainties of the monthly $^{222}$Radon flux normalisation (which is on average 15%). Within the RTM uncertainty there is no disagreement between top-down and bottom-up seasonality. However, our top-down estimated seasonality would also allow a seasonality twice as large as that reported by EDGARv6.0.

[Figure]

Fig. 1: left: Monthly mean normalised $CH_4/^{222}Rn$ slopes of the years 2004-2015 together the mean seasonal variation of EDGARv6.0 emissions of the large Heidelberg catchment area for 2010. Right: mean seasonal cycle of the $CH_4/^{222}Rn$ slopes plotted in the left panel. Error bars correspond to the standard deviation of the individual years and include the uncertainty of the $^{222}Rn$ flux normalisation.

(4) There is no comment anywhere (unless I missed it?) on the bias introduced into the trace gas flux estimations by the fact that only nocturnal measurements are used in this flavour of the RTM. If there is a strong diurnal variation in the fluxes estimated by the nocturnal RTM method, then the results will not be an accurate representation of the diurnal average flux (e.g., $CO_2$ will only deliver respiration fluxes). This should perhaps be noted in the description of the method, along with a justification for why the problem "might not be too bad" for $CH_4$.

The reviewer is correct and we apologise for not having mentioned that all flux estimates refer to nighttime only. We will change this throughout the text.

There is no diurnal cycle of $CH_4$ emissions available for download on the EDGARv6.0 website. But we do have access to diurnal variations of estimated $CH_4$ emissions from the TNO inventory (Kuenen et al., 2021). Weighted by category for the large catchment area we calculated for the time from 21:00 to 4:00h on average about 4% lower emissions compared to the average $CH_4$ flux for the entire day. We will report this number in the revised manuscript.

References:

Karstens, U., Schwingshackl, C., Schmithüsen, D., and Levin, I.: A process-based $^{222}$Radon flux map for Europe and its comparison to long-term observations, Atmos. Chem. Phys., 15, 12845-12865, https://doi.org/10.5194/acp-15-12845-2015, 2015.

Kuenen, J., Dellaert, S., Visschedijk, A., Jalkanen, J.-P., Super, I., and Denier van der Gon, H.: CAMS-REG-v4: a state-of-the-art high-resolution European emission inventory for air quality modelling, Earth Syst. Sci. Data Discuss. [preprint], https://doi.org/10.5194/essd-2021-242, in review, 2021.

---

## Author Response (AR1)

Second reply to comments of Reviewer 1, Alastair Williams

We wish to thank Alastair Williams for his insightful review and helpful suggestions to improve our manuscript, including the English language. First replies to his specific comments #1 - #4 had already been given in our earlier reply posted on October 5, 2021. Here we give our replies to **all** comments and explain the changes we have made in the manuscript.

**Specific Comments**

(1) I believe this manuscript would benefit greatly by clearer elucidation of the role of the "nocturnal accumulation" RTM within the broader context of European top-down trace gas emission estimates using radon. This could be accomplished quite easily by adding short paragraphs in the Introduction, Methods and Discussion sections (suggested locations are provided below under "**Minor and technical comments / suggestions**"), outlining the differences in scope and implementation between the "nocturnal accumulation" (this paper), "tall tower" and "baseline" (mountaintop and remote location) applications of RTM, and emphasizing their complementarity. The "nocturnal accumulation" RTM, applied in the current study, uses surface-based measurements for estimating local fluxes (say, up to 200km spatial scale), and should be contrasted with the RTM as applied to measurements from tall towers, which estimates fluxes up to the regional scale (200-1000km). In the latter case, trace gases are monitored in the deeper mixed / residual layer above the nocturnal inversion and are therefore integrative of the whole boundary layer, the entire diurnal cycle and much bigger fetch areas (regional to continental scale). For these reasons, they are not restricted to nocturnal-only measurements and do not suffer so much from the problem of representing local point sources within the footprint (the strong boundary layer regional mixing process tends to increase the comparability of the trace gas and radon signals). However, they require different assumptions about reference ("background") signals and exchanges with the free troposphere, and have their own special implementation difficulties (e.g., increased uncertainty in the definition of the footprint, losses/gains at the boundaries and the top of the box, non-stationarity due to synoptic weather influences etc.). Finally, RTM applications at baseline stations (mountaintops and remote locations) are similar in implementation to the tall tower case and can be used to estimate fluxes from regional to continental and even hemispheric scales.

We fully agree with the reviewer that $^{222}$Rn in combination with greenhouse gases (GHG) observations has potential to improve top-down estimates of GHG emissions. A review paper on the various applications would certainly be a useful contribution to the scientific literature. It was, however, not our aim to provide such a review. Our focus was only on the "nocturnal accumulation" RTM, also because this is currently the most frequent application of the RTM that reports **quantitative** regional trace and GHGs emissions estimates.

Using larger scale representative $^{222}$Rn observations from coastal, tall tower or mountain stations e.g. for air mass characterisation (marine vs. continental) or for transport model validation (e.g. validating nocturnal boundary layer thickness) is certainly another valuable application of the RTM. However, if such applications shall provide quantitative results, they will also be restricted by the not well known $^{222}$Rn flux from continental soils. In the regional Heidelberg study area, we have the advantage that soil types, their textures and Uranium contents are rather homogeneous. This allowed us to use our long-term $^{222}$Rn flux observations instead of flux model data to estimate RTM-based $CH_4$ emissions. However, a larger footprint (200-1000km) will cover soil types that are much more variable in their $^{222}$Rn flux than in the restricted upper Rhine valley. As the reviewer states correctly, this would probably increase the uncertainty of footprint-weighted RTM-based fluxes considerably. But, no matter what the scale is, local, regional, continental or even hemispheric, in all applications using $^{222}$Rn as a **quantitative** tracer, the big unknown will be the soil exhalation rate. To

our knowledge, there currently exist four different $^{222}$Rn flux maps for Europe developed using different input data and parameterisations. Their long-term medium emission rates differ by a factor of two (see Karstens et al., 2015, Fig. 4). We, thus, only have to hope for a significant improvement of the reliability of soil moisture models, the most important but very variable input parameter to accurately calculate $^{222}$Rn exhalation rates. Note, however, that long-term continuous soil moisture measurements are notoriously difficult to conduct, and we just recently learned that in many cases, they are not precise (besides being not representative) enough to validate modelled soil moistures. At larger scales and in a well-mixed atmospheric boundary layer, the spatial heterogeneity of the $^{222}$Rn, and also of the GHG flux may level out, but the absolute mean $^{222}$Rn flux from the different continents will stay as unknown, with uncertainties of the same order or even larger than the uncertainty of bottom-up GHGs emission inventories (at least in Europe).

We admit that this is a not very promising message. Therefore, we hesitate to further promote the RTM and other $^{222}$Rn applications - for quantitative use - in this study. We rather feel that the Radon community has eventually to face that message and start focussing on extensive $^{222}$Rn flux measurement campaigns in the footprints of atmospheric monitoring stations as well as on developing better soil models (for moisture and for gas transport).

Changes in the manuscript concerning the reviewer's comment are described in our answers given to the specific comments below.

> (2) With regards to the discussions on the effects of point source emissions on the RTM results: If point source emissions are injected directly into the nocturnal inversion layer, or if they are injected above (i.e., from tall stacks) but are then fully or partially incorporated into the inversion layer by subsequent "fumigation" events, then they may be mixed in the footprint of the measurement site and influence the average trace gas levels experienced on a given night. If this is an uncommon occurrence, it will be dismissed as an outlier in the analysis. However, if it happens often, then it may end up being correlated with the radon observations because both scalars are mixed (or partially mixed!) within the same nocturnal volume. In other words, this could lead to a range of scatt in the correlation plots...

We agree with the reviewer and confirm that there were indeed very few occasions, also in the observations, with very high $CH_4/^{222}$Rn slopes, which met the selection criterion and showed stable trajectories with footprints hitting the largest $CH_4$ point source areas in the north-west of Heidelberg (MA/LU). This is discussed in Sec. 3.2.

> (3) Seasonal variations in the radon flux translate to seasonal variations in the measured atmospheric radon concentrations. The latter are only partially matched by corresponding variations in the measured $CH_4$ concentrations, resulting in a residual seasonality in the computed $CH_4/^{222}$Rn ratios. This latter seasonality is initially removed from the ratios, so that a focus can be placed on the effects of the absolute flux errors. The intention appears to be (according to the first paragraph of Section 3.1) that the seasonality in the ratios would be returned to later for separate investigation; however, this is never done.

The reviewer is correct – we were so much overwhelmed by the large variability of the normalised $CH_4/^{222}$Rn slopes, that we decided to only focus on annual mean slopes and corresponding RTM-based $CH_4$ fluxes and finally neglected analysing the results with respect to a possible seasonal variation of the RTM-based $CH_4$ emissions. We now made a respective analysis by calculating normalised monthly mean slopes for the years 2004-2015, i.e. the period when the estimated annual mean fluxes varied from year to year by only 3%. Monthly mean slopes for the individual years, normalised to the long-term mean of all slopes, are plotted in Fig. 1 (left panel), together with the mean seasonal cycle of EDAGARv6.0 $CH_4$ emissions in 2010 from the large catchment area (also shown in the right panel of Fig. 3 of the manuscript). The right panel of Fig. 1 (below) shows the mean seasonal cycle and its standard deviation for 2004-2015. The error bars also include the uncertainties of the monthly $^{222}$Radon flux normalisation (which is on average 15%). Within the RTM

uncertainty there is no disagreement between top-down and bottom-up seasonality. However, our top-down estimated seasonality would also be compatible with a seasonality twice as large as that reported by EDGARv6.0.

[Figure]

Fig. 1: Left: Monthly mean normalised $CH_4/^{222}Rn$ slopes of the years 2004-2015 together with the mean seasonal variation of EDGARv6.0 emissions of the large Heidelberg influence area for 2010. Right: mean seasonal cycle of the $CH_4/^{222}Rn$ slopes plotted in the left panel. Error bars correspond to the standard deviation of the individual years and include the uncertainty of the $^{222}Rn$ flux normalisation.

In the revised manuscript we added a paragraph reporting on the agreement between mean RTM-based and the bottom-up estimated seasonality of $CH_4$ emissions in our footprint, see lines 431 ff.

(4) There is no comment anywhere (unless I missed it?) on the bias introduced into the trace gas flux estimations by the fact that only nocturnal measurements are used in this flavour of the RTM. If there is a strong diurnal variation in the fluxes estimated by the nocturnal RTM method, then the results will not be an accurate representation of the diurnal average flux (e.g., $CO_2$ will only deliver respiration fluxes). This should perhaps be noted in the description of the method, along with a justification for why the problem "might not be too bad" for $CH_4$.

The reviewer is correct and we apologise for not having mentioned explicitly that all flux estimates refer to nighttime only. We have changed this throughout the text of the revised manuscript.

There was no diurnal cycle of $CH_4$ emissions available for download on the EDGARv6.0 website. But we do have access now to diurnal variations of estimated $CH_4$ emissions from the TNO inventory (Kuenen et al., 2021). Weighted by category for the large catchment area we calculated for the time from 21:00 to 4:00h on average about 4% lower emissions compared to the average $CH_4$ flux for the entire day.

We discuss this bias in the revised manuscript in lines 568-570.

**Minor and technical comments / suggestions**

(1) [Abstract] p2, line 16: Here and elsewhere in the manuscript, the authors might consider changing "catchment area" to "flux footprint" or similar. In my experience, the word "catchment" is a hydrological term that refers specifically to an area defined by a watershed (topographical high-altitude line).

The reviewer is correct that the term catchment area comes from hydrology. We therefore decided to use either the term "footprint" or "influence area", the latter mainly when we refer to the emission maps and the large, intermediate or small areas displayed in Fig.1 or Fig. 4 (left panels)

(2) [Abstract] p2, line 18: Change "total $CH_4$ emission" to "total nocturnal $CH_4$ emission".

done

 (3) [Abstract] p2, line 19: Change "exhalation rate from soils" to "exhalation rates estimated from soils".

done

(4) [Abstract] p2, line 23: Change "RTM-based top-down with bottom-up" to "RTM-based top-down flux estimates with bottom-up".

done

(5) [Abstract] p2, line 26: Change "as their emissions are not captured by the RTM method" to "as their emissions may not be fully captured by the RTM method, for example if stack emissions are injected above the top of the nocturnal inversion layer, or if point source emissions from the surface are not well mixed into the footprint of the measurement site".

done

(6) [Introduction] p2, line 37: Change "(UNFCCC, 2015). But only the" to "(UNFCCC, 2015), but only the" OR "(UNFCCC, 2015). However, the".

done

(7) [Introduction] p2, line 41: Change "A possibility to estimate continental" to "A possibility for estimating continental nocturnal".

done

(8) [Introduction] p3, line 45: Change "lifetime of about 5.5 days" to "half-lifetime of about 3.8 days".

done

(9) [Introduction] p3, line 48: After "Liu et al., 1984", you might consider adding "Williams et al., 2011". Reference: Williams AG, Zahorowski W, Chambers SD and Griffiths A, 2011: The vertical distribution of radon in clear and cloudy daytime terrestrial boundary layers. *J Atmos Sci.* **68**:155–174.

We added the suggested reference.

(10) [Introduction] p3, line 50: Change "correlated increases" to "correlated overnight increases".

done

(11) [Introduction] p3, line 52: Change "recommended to use this tracer for transport model validation but also to apply the RTM" to "recommended for use in transport model validation and application of the RTM".

done

(12) [Introduction] p3, end of line 53: This might be a good place to remind the reader that the "nocturnal accumulation" application of the RTM is significantly different from "tall tower" RTM applications. See "**Specific comments**" #1.

We are not sure if a "tall tower" RTM has already been established; at least we are not aware of any publication of such an application.

(13) [Introduction] p3, line 65: Change "when missing precipitation" to "when a lack of precipitation".

done

(14) [Introduction] p3, end of line 68: It would be helpful here to have a short summary of the known effects of increasing near-surface soil moisture on the radon flux. For example, is it a linear / logarithmic relationship, or is it a negligible effect until the soil gets very close to saturation? This would help the reader to get a feel for the potential severity of the problem and prepare them for your discussion of radon fluxes around Heidelberg in later sections.

We refer now already in the Introduction to the Appendix, lines 69-72.

(15) [Introduction] p3, line 73: Remove "and $CH_4$". Otherwise, it is a circular statement ("we use $CH_4$ flux variability to evaluate $CH_4$ emission estimates").

We also studied the influence of the $CH_4$ flux variability. Therefore we did not follow the reviewer's suggestion here.

(16) [Methods 2.1] p4, lines 89-91: Consider enhancing the discussion of H(t) like this: H(t) is a length scale corresponding to the 'effective' depth that the stable layer would have if the trace gases of interest were uniformly mixed vertically within it. The layer is assumed to be mixed 'well enough' that the measured near-surface concentrations can be considered as representative of the layer-averaged values (Williams et al., 2016). Reference: Williams AG, Chambers SD, Conen F, Reimann S, Hill M, Griffiths AD and Crawford J, 2016: Radon as a tracer of atmospheric influences on traffic-related air pollution in a small inland city. *Tellus B* **68**, 30967. http://dx.doi.org/10.3402/tellusb.v68.30967

While we agree that "H(t) is a length scale corresponding to the 'effective' depth that the stable layer would have if the trace gases of interest were uniformly mixed vertically within it", we do not agree that "the layer is assumed to be mixed 'well enough' that the measured near-surface concentrations can be considered as representative of the layer-averaged values".

The largest $^{222}$Rn gradient is observed close to the soil surface, therefore the nocturnal boundary layer is not at all well mixed. What is important to apply the RTM is that $^{222}$Rn and the trace gas are both measured at exactly the same height above ground level and that their vertical gradients are generated by the same process, i.e. turbulent mixing of soil-borne emissions, with the same height-dependent vertical diffusion coefficient K(z) (leading to similar concentration gradients).

We have included the first statement in line 98 ff of the revised manuscript and changed the following text to make our point clearer.

(17) [Methods 2.1] p5, lines 107-108: "... residual layer air that has a $CH_4$/$^{222}$Rn ratio similar to that at the start of the night-time observation period". I assume this is the value you use to define the reference point for $\Delta C$ in the equations above? If so, then maybe mention that here. The encroachment of residual layer air into the growing nocturnal boundary layer is also discussed by Williams et al. (2016): see ref above.

Our approach does not need an explicit reference/background point. We simply correlate half-hourly nighttime $CH_4$ and $^{222}Rn$ data and use the slope of the regressions for the flux estimate. The slope is implicitly the same as if the background concentrations, i.e. at the beginning of the accumulation period, would be subtracted. See also reply on comment (16) and revised text following Eq. (2)

(18) [Methods 2.1] p5, line 113: After "and the trace gas", consider adding ", or at least that surface source functions can be considered to be essentially random and uncorrelated with atmospheric processes operating on short temporal and small spatial scales".

Thank you, this is a very good suggestion, we added corresponding text in line 128-130 of the revised manuscript.

(19) [Methods 2.1] p5, lines 113-120: With regards to the discussions on the effects of point source emissions on the RTM results, you could discuss this further as per "**Specific comments**" #2.

We changed the respective sentence in line 131 to: "… while trace gas plumes from point sources are not or not fully captured..."

(20) [Methods 2.1] p5, line 116: Change "relevant" to "present".

done

(21) [Methods 2.2] p6, line 168: "… this method is only applicable for area sources that are similarly homogeneously distributed as those of $^{222}Rn$ (Eq. 4)". This is true, but see "**Specific comments**" #2.

We still feel that point source contributions are not captured to a significant degree and will therefore not mention the exception described under (19) here again.

(22) [Methods 2.2] p7, line 169: Maybe change "be missing" to "be wholly or partially missing". See "**Specific comments**" #2.

Changed as suggested, see line 186 of the revised manuscript.

(23) [Methods 2.2] p7, lines 174-176: "In the inventories these fluxes are associated with much larger uncertainties than those from point sources and are thus a rewarding target for the RTM". This is an excellent point! See "**Specific comments**" #1.

Thank you, but see also our reply to Specific comments #1 above.

(24) [Methods 2.3] p7, line 179: "The most important pre-requisite to apply the Radon Tracer Method for quantitative GHGs flux estimates are representative $^{222}Rn$ soil exhalation rates in the catchment area". Maybe you should remind the reader here that Eqn (4) implies that errors in the derived GHG fluxes will be directly proportional to errors in the radon fluxes used.

We added in lines 196-197: "as errors in the derived GHG fluxes will be directly proportional to errors in the $^{222}Rn$ fluxes (see Eq. (4))".

(25) [Methods 2.3] p7, line 198: Change "from the sandy soils of M1 and M3" to "from sandy soils (denoted M1 and M3)".

We changed the sentence to "from sandy soils at stations M1 and M3", line 216.

(26) [Methods 2.3] p8, line 206: Change "from M2, M4 and M5" to "from soil types denoted M2, M4 and M5".

We left the text as is because M2, M4 and M5 are names of measurement stations. Their respective soil types are described in the following sentence, lines 224 ff.

(27) [Methods 2.3] p8, lines 216-218: Change "This seasonality… lower right panel of Fig. 4" to "This seasonality in the radon fluxes leads to a seasonal variation in atmospheric radon concentrations which then transfers to the computed $CH_4/^{222}Rn$ ratios because the corresponding $CH_4$ seasonality is relatively small in amplitude. In order to investigate this seasonality separately from the overall effects of the absolute flux errors, the measured and modelled seasonality of $^{222}Rn$ fluxes in the two pixels south of Heidelberg were first normalised to the respective annual means and are shown in the lower right panel of Fig. 4. This will be discussed further in the Results section".

We changed the text with some modifications of the suggested phrasing, see lines 235 ff of the revised manuscript.

(28) [Results 3.1] p10, line 291: After "during all nights", maybe add a description of typical conditions during excluded nights. For example: "Nights excluded by this restriction tended to have higher wind speeds, be cloudy or were disturbed by passing synoptic weather patterns (e.g., fronts)".

We are not able to answer this question as the minority of nights had good correlations (i.e. those with strong nocturnal inversions) while during all other nights all kind of other meteorological conditions prevailed.

(29) [Results 3.1] p10, line 295: Change "the $^{222}Rn$ exhalation rate from soils has a pronounced seasonality" to "as discussed in Section 2.3, the measured and modelled $^{222}Rn$ exhalation rates from soils both exhibit a pronounced seasonality".

done

(30) [Results 3.1] p10, lines 297-298: Change "This seasonality of the $^{222}Rn$ flux imposes a seasonality on the $CH_4/^{222}Rn$ ratios. We therefore normalised…" to "This seasonality of the $^{222}Rn$ flux results in a seasonality in atmospheric radon concentrations and consequently also the computed $CH_4/^{222}Rn$ ratios (as the corresponding $CH_4$ seasonality is relatively small in amplitude). In the analysis to follow, we first normalised...".

done

(31) [Results 3.1] p11, lines 297-299: Change "to the annual mean $^{222}Rn$ flux" to "that adjusts the $^{222}Rn$ flux to its annual mean value".

done

(32) [Results 3.1] p11, lines 300-301: Change "This intermediate step was taken because of the large uncertainty of the *absolute* $^{222}Rn$ flux in contrast to its much better defined seasonality" to "This intermediate step was taken in order to separately study both the large uncertainty of the *absolute* $^{222}Rn$ flux and its much better defined seasonality". See "**Specific comments**" #3.

We did not change the sentence as suggested, but we included here a remark (line 331) that the seasonality we found in the $CH_4/^{222}Rn$ slopes (Fig. 1 above) will be discussed later i.e. in Sec. 3.5 of the revised manuscript.

(33) [Results 3.1] p11, lines 325-328: "As mentioned … afternoon before". It would be really nice to see an illustration of this by showing examples of $^{222}Rn$ and $CH_4$ hourly time series for two

contrasting nights characterized by strong positive and strong negative correlations. In the latter case, is the computed equivalent mixing layer depth H close to 30m?

As was explained above (see remark 16), we are not able to determine H(t) (it is only a "virtual height"). Also we excluded nights with **negative** correlations. We only included nights where both tracers decreased, but were still positively correlated. These were synoptic variations, e.g. a change from continental to marine air.

(34) [Results 3.2] p12, lines 334-336: Change "The $CH_4/^{222}Rn$ slopes displayed … in the footprint of Heidelberg" to "The $CH_4/^{222}Rn$ slopes displayed in Fig. 5 show large variability. It is of interest to explore if this variability can be explained by spatial variations in the $CH_4$ emissions, and if so, the extent to which we can associate the high-slope cases with hot spot emission areas in the footprint of Heidelberg".

We changed the sentences as suggested, see lines 365 ff of the revised manuscript.

(35) [Results 3.2] p12, line 336: Change "air mass influence" to "air mass footprint".

done

(36) [Results 3.2] p12, line 338: Change "origin is from" to "has passed over".

done

(37) [Results 3.2] p12, line 358: Change "will not be captured" to "may not be fully captured". See my previous comments.

done

(38) [Results 3.2] p12, line 360: Change "can we" to "we can".

done

(39) [Results 3.4] p13, line 384: Change "M2, M4 and M5 to" to "M2, M4 and M5 to be".

done

(40) [Results 3.4] p13, line 385: Change "The corresponding $CH_4$ flux it is plotted as" to "The corresponding calculated $CH_4$ flux is plotted as the".

done

(41) [Discussion 4.1] p16, line 483: Change "captured" to "fully captured".

done

(42) [Discussion 4.2] p18, line 543-544: Change "could show" to "have shown".

done

(43) [Discussion 4.2] p18, line 545: Change "ask for" to "dictate a need for".

done

(44) [Discussion 4.2] p18, lines 546-550: "A second problem … less well-defined $^{222}$Rn fluxes". I think a slightly more detailed discussion is needed here. See my suggestions in "**Specific comments**" #1.

As explained in our reply to Specific Comment #1 and as mentioned again in the reply to comment (12), we did not make any other changes to the text than adding "nocturnal accumulation" in line 594 of the revised manuscript.

(45) [Conclusions 5] p19, line 583: Change "quantitative flux estimation relies on the accuracy" to "quantitative flux estimation relies critically on the accuracy".

done

References:

Karstens, U., Schwingshackl, C., Schmithüsen, D., and Levin, I.: A process-based $^{222}$Radon flux map for Europe and its comparison to long-term observations, Atmos. Chem. Phys., 15, 12845-12865, https://doi.org/10.5194/acp-15-12845-2015, 2015.

Kuenen, J., Dellaert, S., Visschedijk, A., Jalkanen, J.-P., Super, I., and Denier van der Gon, H.: CAMS-REG-v4: a state-of-the-art high-resolution European emission inventory for air quality modelling, Earth Syst. Sci. Data Discuss. [preprint], https://doi.org/10.5194/essd-2021-242, in review, 2021.

**Reply to comments of Reviewer 2, Claudia Grossi**

We wish to thank Claudia Grossi for her careful review and her suggestions to improve our manuscript. Our replies to her comments and changes we made in the revised manuscript are printed in blue.

**Specific comments**

**Section: 1. Introduction**

- In Lines 55-56 authors declare that RTM has been applied assuming a spatially homogenous radon flux (bibliography here stops to 2009 and they could also add Vogel et al., 2012 Wada et al., 2013 and Grossi et al., 2014). Furthermore, this sentence is not fully correct because Grossi et al., 2018 applied the RTM calculating the effective radon flux. This was calculated by coupling radon flux data, obtained using the output for the 40-year climatology obtained with the model developed by López-Coto et al. (2013), with the footprints calculated by the ECMWF-FLEXPART model (version 9.02) (Stohl, 1998) (For more info please look at the Figure 8 of Grossi et al., 2018 and equation n. 3). In addition, in Grossi et al., 2018 the $CH_4$ fluxes, obtained by RTM, were also compared for the first time with $CH_4$ fluxes obtained coupling the EDGAR inventory with ECMWF-FLEXPART footprints for the same period.

Thank you for reminding us on these references. We apologise for the omission, included the most important references in the revised manuscript and changed the text in lines 59 ff (subm. ms) to:

"… (Levin, 1984; Gaudry et al., 1990; Levin et al., 1999; 2011; Biraud et al., 2000; Schmidt et al., 2001; Hammer and Levin, 2009; Vogel et al., 2012; Wada et al., 2013; Grossi et al., 2018). In most of the earlier studies the $^{222}$Rn flux from the soil has been assumed as spatially homogeneous and varying only slightly on the seasonal time scale. Recent research has, however, challenged this perception of a homogeneous and temporally almost constant flux. Several attempts to model $^{222}$Rn exhalation rates from European soils revealed rather large spatial variability (Szegvary et al., 2009; Lopez-Coto et al., 2013; Karstens et al., 2015). Therefore, Grossi et al. (2018) applied the RTM by calculating the effective $^{222}$Rn flux influencing their station by coupling the flux map from Lopez-Coto et al. (2013) with atmospheric transport model calculated footprints."

- In Lines 66-67 authors say that the basic assumption for the classical RTM application is of having a more or less constant radon flux. I do not personally agree with this and I think it already stays in the past. Nowadays it is known that for correctly applying the RTM we need high quality and reliable atmospheric GHGs and radon concentration data and validated radon flux models with as high as possible spatial and temporal resolution. These are actually between the main goals of the project EMPIR traceRadon (presented in Röttger et al., 2021) which wants to offer also a metrology for radon flux measurements and sensitivity studies for the RTM applications.

I think it will be nice to have all this previous information in the state of the art.

We agree to Claudia Grossi and now refer to the EMPIR traceRadon project by citing in our Methods and Discussion sections the paper by Röttger et al. (2021), which has only been published after submission of our manuscript. However, to our understanding this paper does not present any new findings regarding the RTM.

**Section: 2. Methods**

*Radon flux estimation for the RTM application*: The structure of this section does not help the reader to understand the methodology applied for the calculation of the radon flux used within the RTM. I had the impression that authors finally used a constant value of 18.3 ± 4.7 mBq m$_{-2}$ s$_{-1}$. Is it correct? Did you not estimated the effective flux seen by the station coupling radon flux climatology output from Karstens et al., 2015 with STILT footprints? It may help to have this info in a dedicated paragraph where the estimation of the radon flux used for the RTM is clearly explained.

For the discussion of model footprint-weighted $^{222}$Rn fluxes, it is helpful to emphasise the fundamental difference between the Grossi et al. (2018) approach and the approach in this manuscript. Due to the well-documented deficiencies of transport models to simulate the nocturnal boundary layer (e.g. Geels et al., 2007; Gerbig et al., 2008), we do NOT base our $^{222}$Rn flux estimates on nocturnal footprint modelling. We rather indeed used the **mean** observation-based $^{222}$Rn flux value of 18.3 ± 4.7 mBq m$^{-2}$ s$^{-1}$ to estimate mean CH$_4$ emissions in the Heidelberg footprint (as presented in Fig. 7 of our manuscript). Principally the same is true for the STILT simulation results (Fig. 8) when applying the RTM on modelled CH$_4$ and $^{222}$Rn concentrations. But to be comparable with the observations, we "adjusted" the model-based RTM results by the factor 18.3/16.7 to take into account that the STILT runs were conducted with a slightly lower mean $^{222}$Rn flux, i.e. with the mean of the two flux maps reported in Karstens et al. (2015) (see lines 483 ff of the revised manuscript).

Therefore, in both cases we applied the RTM exactly in the way described in our Sec. 2.1.

As explained in the manuscript, we think that this fundamentally different approach to Grossi et al. (2018), is - as much as possible - independent from model deficits to simulate nighttime situations, particularly during strong inversions.

References:

Geels, C., Gloor, M., Ciais, P., Bousquet, P., Peylin, P., Vermeulen, A. T., Dargaville, R., Aalto, T., Brandt, J., Christensen, J. H., Frohn, L. M., Haszpra, L., Karstens, U., Rödenbeck, C., Ramonet, M., Carboni, G., and Santaguida, R.: Comparing atmospheric transport models for future regional inversions over Europe – Part 1: mapping the atmospheric CO$_2$ signals, Atmos. Chem. Phys., 7, 3461–3479, https://doi.org/10.5194/acp-7-3461-2007, 2007.

Gerbig, C., Körner, S., and Lin, J. C.: Vertical mixing in atmospheric tracer transport models: error characterization and propagation, Atmos. Chem. Phys., 8, 591–602, https://doi.org/10.5194/acp-8-591-2008, 2008.

*STILT footprints*: It will help to have a dedicated section where the calculation of the STILT footprint is described. How long were the back trajectories used for it? Which was the height of the boundary layer used in the STILT simulations for it? I was not able to find this info in the manuscript and it could be useful, as explained in Grossi et al., 2018, when effective radon flux is estimated using also model footprints and for future RTM applications protocols.

We do not think it is important to explain here in detail the STILT footprint modelling as it was used only to roughly estimate the possible catchment area (cf. Fig. 2) or for the routine simulations at the ICOS Carbon Portal described elsewhere. Deficits in boundary layer height estimates largely cancel by applying the RTM also to the simulated concentrations (see earlier comment).

*CH$_4$ and $^{222}$Rn measurements*: I agree with the authors on the importance of correctly estimating the radon flux values used in the RTM but equation 3 gives the same weight to methane and radon concentration measurements too. I think it will be nice to have a paragraph dedicated to experimental measurements (CH$_4$, $^{222}$Rn and meteorology). Here, a 24-year dataset of radon progeny ($_{214}$Po) concentration measured using a static filter method (Levin et al., 2002), and a constant disequilibrium correction factor between $^{222}$Rn and $_{214}$Po of 1.11 (F$_e$), has been used as explained in lines 275-282. However, results from inter-comparison studies between radon and radon progeny monitors based on different measurements techniques (Grossi et al., 2016; Schmithüsen et al., 2017; Grossi et al. 2020) show that under saturated atmospheric water conditions and low atmospheric aerosol concentration this disequilibrium factor F$_e$ could change inducing an underestimation of atmospheric radon activity concentration by static filter methods. In addition, Levin et al., 2017 estimated a correction factor to take into account the $_{222}$Rn progeny loss in long tubing based on static filter measurements in the laboratory and in the field. Taking into account all these previous outcomes, I wonder if authors filtered they data for rain/low aerosol episodes and/or have applied these corrections factors before using the dataset for RTM calculation. It should be clearly stated in the manuscript. Finally, authors say (Line 279) that they used an ANSTO scale to calibrate their instrument. Unfortunately ANSTO instruments, running at several ICOS stations, are calibrated using their own source which is located within each instrument. This means that there is not a primary standard or a second transfer standard instrument to harmonize these instruments and using a ANSTO scale does sound correct to be used. The lack of a ambient radon measurement metrology was one of the main aims of the traceRadon project. For example, in Grossi et al., 2020 the correlation of the same ARMON (Grossi et al., 2012) with two ANSTO monitors (located respectively at Saclay (100 m.a.g.l.) and at Orme de Marisier (5 m.a.g.l.)) had slopes of 0.97 ± 0.01 and 0.96 ± 0.01 with intercepts of 0.01 ± 0.06 and 0.01 ± 0.06, respectively. Schmithüsen et al., 2017 found a correlation, at the Heidelberg station, between the HRM and the ANSTO instrument of 1.22 ± 0.01 with an intercept of 0.42 ± 0.04.

The reviewer is correct in pointing out possible errors in the $^{222}$Rn observations conducted with the HRM via the measurement of its progeny $^{214}$Po. For the Heidelberg measurements we currently assume a constant disequilibrium between $^{222}$Rn and its measured progeny $^{214}$Po, which in reality is not the case. The combined mean comparison factor of ANSTO/HRM = 1.22 published by Schmithüsen et al. (2017) is based on data from a long-term comparison campaign between the HRM and an ANSTO detector conducted at the Heidelberg measurement site. Here the air intake of the HRM was always through a <2m tubing, so that we did not apply a correction for aerosol loss in tubing. However, we agree that there may still be adjustments to be applied to the ANSTO/HRM factor we used here to estimate ambient $^{222}$Rn at the Heidelberg site since 1996. Our currently best estimate of his factor and the corresponding uncertainty of half-hourly values, i.e. less than 15% (1 sigma), includes such possible adjustments (e.g. the ANSTO data generally show higher values at low activity concentrations than the HRM, which may be due to a not yet applied deconvolution of the ANSTO data and requires further investigation).

In the revised manuscript we added a few sentences noting that the published factor 1.22 may be subject to future changes, once we have new intercomparison and calibration results available. We cite here the paper by Röttger et al. (2021) that describes the planned future calibration and comparison work e.g. in the traceRadon project. See lines 298 ff of the revised manuscript.

**Section 3: Results**

*Paragraph 3.5*: The comparison of STILT based and RTM based results of the CH$_4$/$^{222}$Rn slopes is obtained, if I understood correctly, comparing the ratios between CH$_4$ and $^{222}$Rn

concentrations simulated using the STILT model in forward mode and using, as emissions, the radon flux climatology output from the model presented in Karstens et al., 2015 with the ratios of measured $CH_4/^{222}Rn$. Is it correct? Authors said that these comparison seems to work properly. May this due to the fact that RTM is used only applying a constant radon flux values over the time and area where here the forward simulation is run with spatial radon flux climatology?

By calculating $CH_4/^{222}Rn$ slopes from STILT-simulated concentrations we wanted to be as close to reality as possible (see explanations above). In the real world the nocturnal slopes (shown in Fig. 5 of the manuscript) are influenced by the heterogeneous $^{222}Rn$ flux in our (changing) footprint, while for the model we use the heterogeneous fluxes from the modelled maps of Karstens et al. (2015). Similarly, for observations and model results, when finally estimating annual mean $CH_4$ emissions we used the average of the corresponding heterogeneous $^{222}Rn$ fluxes. With this we came up with similar mean $CH_4$ emissions. By this we confirm the mean flux from the EDAGARv6.0 emissions inventory, although we believe that the observational RTM results underestimate the real flux in our footprints as they largely miss the emissions from point sources (that are partly taken into account in the STILT-based RTM results because the model resolution was too coarse).

*Discussion and Conclusion*: Authors may revisit the discussion and conclusions sections taking into account the previous comments expressed by the reviewer.

We added the most important conclusions from the above in the respective sections of the revised manuscript.

**Minor and technical comments / suggestions**

- Please use radon or $^{222}Rn$ instead of $^{222}Radon$ within the manuscript. The same nomenclature for $^{226}Radium$ and $^{214}Polonium$.

We use the term $^{222}Rn$ throughout the manuscript and $^{222}Radon$ only when it shows up for the first time.

- Line 384 mBq instead of Bq.

Thanks for spotting this mistake!

**References to be included**

Grossi, C., Arnold, D., Adame, A. J., Lopez-Coto, I., Bolivar, J. P., de la Morena, B. A., and Vargas, A.: Atmospheric 222Rn concentration and source term at El Arenosillo 100 m meteorological tower in southwest Spain, Radiat. Meas., 47, 149–162, https://doi.org/10.1016/j.radmeas.2011.11.006, 2012.

Grossi, C., Àgueda, A., Vogel, F. R., Vargas, A., Zimnoch, M., Wach, P., Martín, J. E., López-Coto, I., Bolívar, J. P., Morguí, J.-A., and Rodó, X.: Analysis of ground-based 222Rn measurements over Spain: filling the gap in southwestern Europe, J. Geophys. Res.-Atmos., 121, 11021–11037, https://doi.org/10.1002/2016JD025196, 2016.

Grossi, C., Vogel, F. R., Curcoll, R., Àgueda, A., Vargas, A., Rodó, X., and Morguí, J.-A.: Study of the daily and seasonal atmospheric CH4 mixing ratio variability in a rural Spanish region using 222Rn tracer, Atmos. Chem. Phys., 18, 5847–5860, https://doi.org/10.5194/acp-18-5847-2018, 2018

Grossi, C., Chambers, S. D., Llido, O., Vogel, F. R., Kazan, V., Capuana, A., Werczynski, S., Curcoll, R., Delmotte, M., Vargas, A., Morguí, J.-A., Levin, I., and Ramonet, M.: Intercomparison study of atmospheric 222Rn and 222Rn progeny monitors, Atmos. Meas. Tech., 13, 2241–2255, https://doi.org/10.5194/amt-13-2241-2020, 2020.

Röttger et al 2021 Meas. Sci. Technol. in press https://doi.org/10.1088/1361-6501/ac298d

Vogel, F. R., Ishizawa, M., Chan, E., Chan, D., Hammer, S., Levin, I., and Worthy, D. E. J.: Regional non-CO2 greenhouse gas fluxes inferred from atmospheric measurements in Ontario, Canada, J. Integr. Environ. Sci., 9, 45–55, https://doi.org/10.1080/1943815X.2012.691884, 2012.

Wada, A., Matsueda, H., Murayama, S., Taguchi, S., Hirao, S., Yamazawa, H., Moriizumi, J., Tsuboi, K., Niwa, Y., and Sawa, Y.: Quantification of emission estimates of CO2, CH4 and CO for East Asia derived from atmospheric radon-222 measurements over the western North Pacific, Tellus B, 65, 18037, https://doi.org/10.3402/tellusb.v65i0.18037, 2013.

The references from this list, which we included in the revised manuscript are mentioned in our replies above.